# Molecular and electronic structure of terminal and alkali metal-capped uranium(V) nitride complexes

David M. King[1], Peter A. Cleaves[2], Ashley J. Wooles[2], Benedict M. Gardner[2], Nicholas F. Chilton[2], Floriana Tuna[3], William Lewis[1], Eric J.L. McInnes[3] & Stephen T. Liddle[2]

Determining the electronic structure of actinide complexes is intrinsically challenging because inter-electronic repulsion, crystal field, and spin–orbit coupling effects can be of similar magnitude. Moreover, such efforts have been hampered by the lack of structurally analogous families of complexes to study. Here we report an improved method to U≡N triple bonds, and assemble a family of uranium(V) nitrides. Along with an isoelectronic oxo, we quantify the electronic structure of this $5f^1$ family by magnetometry, optical and electron paramagnetic resonance (EPR) spectroscopies and modelling. Thus, we define the relative importance of the spin–orbit and crystal field interactions, and explain the experimentally observed different ground states. We find optical absorption linewidths give a potential tool to identify spin–orbit coupled states, and show measurement of $U^V \cdots U^V$ super-exchange coupling in dimers by EPR. We show that observed slow magnetic relaxation occurs via two-phonon processes, with no obvious correlation to the crystal field.

[1] School of Chemistry, University of Nottingham, University Park, Nottingham NG7 2RD, UK. [2] School of Chemistry, University of Manchester, Oxford Road, Manchester M13 9PL, UK. [3] School of Chemistry and Photon Science Institute, University of Manchester, Oxford Road, Manchester M13 9PL, UK. Correspondence and requests for materials should be addressed to E.J.L.M. (email: eric.mcinnes@manchester.ac.uk) or to S.T.L. (email: steve.liddle@manchester.ac.uk).

In this nuclear age, there is a need for improved extraction agents for separations and recycling technologies[1]. To achieve this, we need a better understanding of the electronic structure of actinide complexes. This is challenging because relativistic effects are important[2]. Furthermore, in contrast to lanthanide and transition metal ions, after inter-electronic repulsion, crystal field (CF) effects induced by coordinated ligands and the spin–orbit coupling (SOC) of such heavy elements can be large and of the same order of magnitude[3]. Thus, neither traditional LS–Russell–Saunders nor $jj$-coupling are entirely appropriate schemes to describe the electronic structures of $5f^n$ elements. However, this is precisely the area where accurate models are needed because our understanding of core chemical concepts such as oxidation state, valence and the spin ($S$), orbital ($L$), and total angular momentum ($J$) quantum numbers become increasingly nebulous at the foot of the periodic table.

In recent years, many novel uranium molecules have become available, some accompanied by detailed spectroscopic and magnetic data[4–10]. However, a comprehensive electronic structure picture has thus far been limited to a few isolated studies of highly symmetric species[5–8,11,12]. In this regard, $5f^1$ complexes are appealing[11,12] because inter-electronic repulsion is absent. Hence, it would be ideal to study systematic families of uranium(V) complexes, but this chemistry has lagged behind that of uranium(IV/VI) because of the propensity of the former to disproportionate into the latter pair under aqueous conditions. Non-aqueous approaches allow isolation of stable uranium(V)[12,13]. For example, the mono(oxo) complex [U(Tren$^{TIPS}$)(O)] [1, Tren$^{TIPS}$ = N(CH$_2$CH$_2$NSiPr$^i_3$)$_3$] exploits the bulky Tren$^{TIPS}$ ligand to block disproportionation and stabilizes a wide range of novel uranium–ligand linkages[14–21].

In 2012, we reported a molecular terminal uranium(V) nitride[20–23], via oxidation of [U$^{III}$(Tren$^{TIPS}$)] (2) by sodium azide to give the dinuclear contact ion pair (DCIP) complex [{U$^V$(Tren$^{TIPS}$)($\mu$-N)($\mu$-Na)}$_2$] (3Na). Treatment of 3Na with 12-crown-4 ether (12C4) afforded the terminal nitride separated ion pair (SIP) complex [U$^V$(Tren$^{TIPS}$)(N)][Na(12C4)$_2$] (4Na)[20]; when the crown was mismatched to the sodium, the contact ion pair (CIP) complex [U$^V$(Tren$^{TIPS}$)($\mu$-N){Na(15C5)}] [5Na, 15C5 = 15-crown-5 ether] was formed[21]. Complex 4Na could also be oxidized to [U$^{VI}$(Tren$^{TIPS}$)(N)] (6)[21]. Terminal uranium nitrides were without precedent outside of matrix isolation experiments[22,24–29]; molecular species always exhibited bridging nitrides[30–41] or decomposed via C–H activation to give amides[21,42]. These difficulties stand in contrast to the common terminal uranium oxides[9,14,43–45] and transition metal nitrides[46,47].

These [U(Tren$^{TIPS}$)(X)]$^{n-}$ complexes, in addition to their synthetic utility[13,48], are appealing because of their axial ($C_3$) symmetry and the availability of a series in which only X and/or counter-ions vary. For example, we can study the influence of varying CF at uranium(V) by replacing O$^{2-}$ with N$^{3-}$ in isostructural and isoelectronic molecules: there are no other available uranium compound families where this is possible. However, extending the nitride family requires more reliable synthetic routes because, although the synthesis of 3Na is conceptually simple, it is not straightforward and sometimes fails for no obvious reason.

Herein, we report a reliable route, enabling us to prepare analogues with all the alkali metals (M) from Li–Cs; the nitride is capped by two bridging M in DCIP derivatives (3M), capped by one M in CIP derivatives (5M) or is terminal in a SIP (4M). We present experimental magnetic/electron paramagnetic resonance (EPR) data that are sensitive to the nature and ordering of the lowest lying electronic states, and ultraviolet/visible/near-infrared (UV/Vis/NIR) data that define the energies of excited states. We have performed *ab initio* calculations on model and real complexes, and parameterize the results in a CF + SOC model, or, in other words, into a relatively simple conceptual basis familiar to chemists. Finally, we refine the model parameters by global fitting to the experimental data to provide a quantitative electronic structure picture for these uranium(V) nitrides and the isoelectronic oxo analogue 1, and justify the experimentally observed differences between these species. Furthermore, we explore the dynamic magnetic properties of the entire series, demonstrating that they are behaving as single molecule magnets (SMMs), and we explore the mechanism for this effect.

## Results

**Synthesis.** Treatment of [U(Tren$^{TIPS}$)(N$_3$)] (7) (ref. 21) with lithium or sodium powders, potassium graphite (KC$_8$), or rubidium or caesium metals in benzene or toluene produces powders, from which 3Li, 3Na, 3K, 3Rb and 3Cs DCIP complexes [{U(Tren$^{TIPS}$)($\mu$-N)($\mu$-M)}$_2$] are consistently isolated in 35–57% crystalline yields (Fig. 1). The alkali metals are straightforwardly abstracted from 3M using sized-matched crown ethers to give the SIP complexes [U(Tren$^{TIPS}$)(N)][M(crown)$_2$] (4M): thus, 12C4 gives 4Na, 15C5 gives 4K and benzo-15-crown-5 (B15C5) gives 4Rb and 4Cs. Abstractions proceed quantitatively to give 4M as oils, with varying crystalline yields (21–95%). However, we were not able to prepare 4Li from 3Li with any abstracting ligand attempted (Supplementary Methods). We find that the entire CIP series [U(Tren$^{TIPS}$)($\mu$-N){M(crown)}] (5M) can be made from 3M, in crystalline yields of 34–66%, by mismatching the crown to alkali metal, using variously 12C4 (5Li), 15C5 (5Na), dibenzo-18-crown-6 (DB18C6; 5K) and 18-crown-6 (18C6; 5Rb and 5Cs). We note the synthetic utility of 4M, all of which can be oxidized with iodine to give 6 in crystalline yields of ∼60% (Supplementary Methods).

**Structural characterization.** Single crystal X-ray diffraction studies on 3Li–3Cs, 4Na–4Cs and 5Li–5Cs (3Na, 4Na, 5Na and 3K were determined previously[20,21,23]; here we report a new polymorph of 3K) show, in each case, the uranium ion coordinated to a nitride and a Tren$^{TIPS}$ in a distorted trigonal–bipyramidal geometry (Figs 2 and 3), with typical U–N$_{amide}$ (∼2.3 Å) and U–N$_{amine}$ (∼2.7 Å) bond lengths (Supplementary Data 1).[49] The DCIP series 3M have {UN(M)$_2$NU} cores, varying from planar for 3Li, 3Na and 3Cs to slightly bent-*trans* nitride geometries for 3K ($\Sigma\angle$ = 350.8(3)°) and 3Rb ($\Sigma\angle$ = 354.79 (16)°). The U≡N distances for 3Li–3K (1.883 (4)–1.929(6) Å) are longer than for 3Rb and 3Cs (1.846(3) and 1.860(5) Å, respectively; though 3Cs and 3Li,3Na just overlap statistically). There are agostic-type interactions between M and *iso*-propyl C–H bonds. For the SIP complexes 4M, there are no significant interactions between the [U(Tren$^{TIPS}$)(N)]$^-$ anion and M, and thus the anions have statistically indistinguishable U≡N distances (1.801(7)–1.825(15) Å, consistent with U≡N triple bonds[50]). The CIP complexes 5M have U–N–M cores, where M is capped by a crown ether. The U≡N distances vary from 1.803(5) to 1.840(3) Å, reflecting varying size and polarizing power of M. The U–N–M angles vary from linear for 5Na and 5Rb, residing on crystallographic threefold axes, to bent for 5Li, 5K and 5Cs (172.1(5), 162.76(16) and 148.8(3)°, respectively).

**Electronic structure.** To set up the quantitative analysis below, we first build a conceptual framework for the uranium nitride group, then develop this into *ab initio* models, and finally parameterize the results to a CF and SOC Hamiltonian.

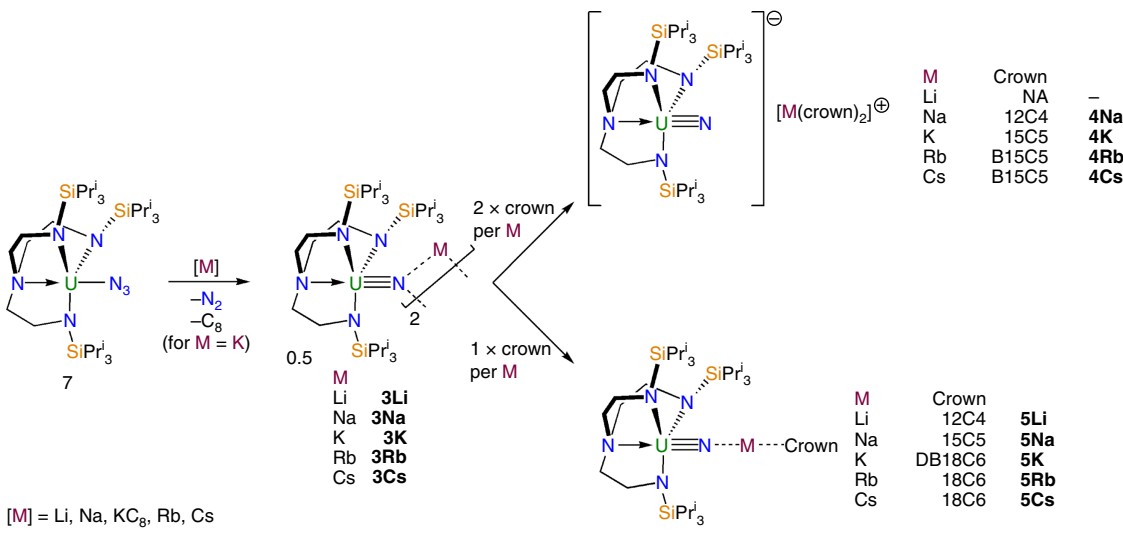

**Figure 1 | Synthetic routes to 3M–5M.** Synthesis of the DCIP series **3M** (M = Li, Na, K, Rb, Cs) from **7** and subsequent derivatization of **3M** to SIP series **4M** (M = Na, K, Rb, Cs) and CIP series **5M** (M = Li, Na, K, Rb, Cs). Crown ether key: 12C4 = 12-crown-4; 15C5 = 15-crown-5; B15C5 = benzo-15-crown-5; 18C6 = 18-crown-6; DB18C6 = dibenzo-18-crown-6. The hypothetical **4Li** has proven inaccessible despite the treatment of **3Li** with a wide range of potential co-ligands.

The uranium(V) free-ion has a $5f^1$ ground state with a SOC constant of $\lambda \sim 2,174\,\mathrm{cm}^{-1}$ (ref. 51), putting the excited $^2F_{7/2}$ SO-coupled multiplet $\sim 7,600\,\mathrm{cm}^{-1}$ above $^2F_{5/2}$. For $f^1$, there is no inter-electronic repulsion, and the Russell–Saunders and $jj$-coupling schemes are equivalent. The dominant perturbation to the free ion is the formal triple bond to $N^{3-}$. An idealized molecular orbital (MO) model of a $[UN]^{2+}$ species consists of the following: a $\sigma$-bond between a filled $2sp$-hybridized nitride orbital and the metal-based $l_z = 0$ ($5f_\sigma$) orbital; two $\pi$-bonds between the filled nitride $2p$-orbitals and the uranium $l_z = \pm 1$ ($5f_\pi$) orbitals; a pair of non-bonding $l_z = \pm 3$ ($5f_\varphi$) uranium orbitals containing one electron; two non-bonding empty $l_z = \pm 2$ ($5f_\delta$) orbitals; two empty $\pi^\star$; and one empty $\sigma^\star$ anti-bonding orbitals (Supplementary Fig. 1). The $\pi < \sigma$ MO ordering, owing to partial anti-bonding character in the $\sigma$-MO arising from the annular lobes of the $5f_\sigma$ orbital, is a known feature of short uranium–ligand multiple bonds[52]. This simple model neglects U($5f/6d$; up to 15% $6d$) and U($5f$)/N($2sp$) mixing but, despite this, the non-bonding ($l_z = \pm 3, \pm 2$) and anti-bonding orbitals have predominantly metal character (>80%). Following the approach of Eisenstein and Pryce[53,54], we consider solely these 'metal-based' orbitals to be represented by the $^2F$ ($s = 1/2$, $l = 3$) basis, where $l_z = \pm 1$ and 0 correspond to the $\pi^\star$ and $\sigma^\star$ orbitals, respectively. The equatorial electron density for each orbital is directly related to the $l_z$ quantum number, therefore they order as $E_{\pm 3} < E_{\pm 2} < E_{\pm 1} < E_0$.

To estimate these energies, we performed Complete Active Space Self-Consistent Field *ab initio* calculations on the hypothetical $[UN]^{2+}$ ion with the average experimental bond length of 1.84 Å. With a minimal active space of one electron in the $5f$-orbitals the seven configuration state functions directly represent the relative orbital energies. Setting $E_{\pm 3} = 0$ gives $E_{\pm 2} \sim 1,300$, $E_{\pm 1} \sim 4,800$ and $E_0 \sim 6,300\,\mathrm{cm}^{-1}$ (Figs 4 and 5a; Supplementary Table 1). Including SOC, which couples the spin $s$ and orbital $l$ angular momenta to give the total angular momentum $j$, the resulting 14 states are paired as seven Kramers doublets, labelled by the $j_z$ quantum number given the uniaxial symmetry (Fig. 4). The calculations give a $j_z = \pm 5/2$ ground state that is well isolated from the $\pm 3/2$ first excited state ($\sim 2,000\,\mathrm{cm}^{-1}$; Supplementary

Table 2). The effective $g_z$ value of the ground doublet (employing the effective $s = 1/2$ formalism) is calculated to be 4.20 ($g_{x,y} = 0$).

We can parameterize the *ab initio* results to a CF + SOC model with the Hamiltonian $\hat{H}_{\mathrm{CF(ax)+SO}} = B_2'^0 \hat{O}_2^0 + B_4'^0 \hat{O}_4^0 + B_6'^0 \hat{O}_6^0 + \lambda \hat{l} \cdot \hat{s}$, having uniaxial symmetry appropriate for the $C_{\infty v}$ point group of $[UN]^{2+}$, where the $\hat{O}_k^q$ operators are the extended Stevens operator equivalents comprising polynomials of the orbital angular momentum operator $(\hat{l})^{3,55}$. We note that this is an effective CF parameterization and thus the Stevens operator equivalent factors are subsumed in the $B_k'^q$ parameters (see Methods, Crystal field Hamiltonian and fitting strategy). A set of simultaneous equations (Supplementary Equations 1–3; Supplementary Note 1) yields the $B_k'^0$ parameters (Supplementary Table 3) from the orbital energies, and fitting the *ab initio* SOC-doublet energies gives $\lambda = 2,000\,\mathrm{cm}^{-1}$. The ground doublet has $g_z = 4.21$, which lies in between the SOC $\gg$ CF limit of $|j, j_z\rangle = |5/2, \pm 5/2\rangle$ ($g_z = 4.29$) and the CF $\gg$ SOC limit of $|l_z, s_z, j_z\rangle = |\pm 3, \mp 1/2, \pm 5/2\rangle$ ($g_z = 4.00$).

After the nitride, the next strongest perturbation in **3M–5M** is the trigonal Tren$^{\mathrm{TIPS}}$ ligand. As a first model, we performed *ab initio* calculations on the $C_3$-symmetrized $[U(NH_3)(NH_2)_3(N)]^-$ that mimics the amine and amide donors (Fig. 5b, Supplementary Fig. 2; Supplementary Table 4). There are three major effects on the $f$-orbitals compared with $[UN]^{2+}$: (i) an increase in the total spread of energies; (ii) the $l_z = \pm 3$ pair is split substantially (by $\sim 1,100\,\mathrm{cm}^{-1}$); (iii) the $l_z = \pm 2$ pair (which remains degenerate) falls below the barycentre of the $|l_z| = 3$ pair. The first effect is due to the additional axial ligand *trans* to the nitride, the second effect is due to the threefold equatorial field mixing the $l_z = \pm 3$ orbitals, and the last effect is due to the equatorial field destabilizing the in-plane $l_z = \pm 3$ orbitals. Thus, when SOC is introduced (Supplementary Table 5): (i) the first excited doublet (still $j_z \approx \pm 3/2$) is at much lower energy than for $[UN]^{2+}$ ($\sim 300\,\mathrm{cm}^{-1}$; all other doublets at much higher energy) and (ii) $g_z$ of the ground doublet (still $j_z \approx \pm 5/2$) is reduced to 3.80 and $g_{x,y}$ is no-longer zero. Due to the loss of $C_{\infty v}$ symmetry, $j_z$ is not a good quantum number; however, owing to the strong axial potential, we use them as approximate labels.

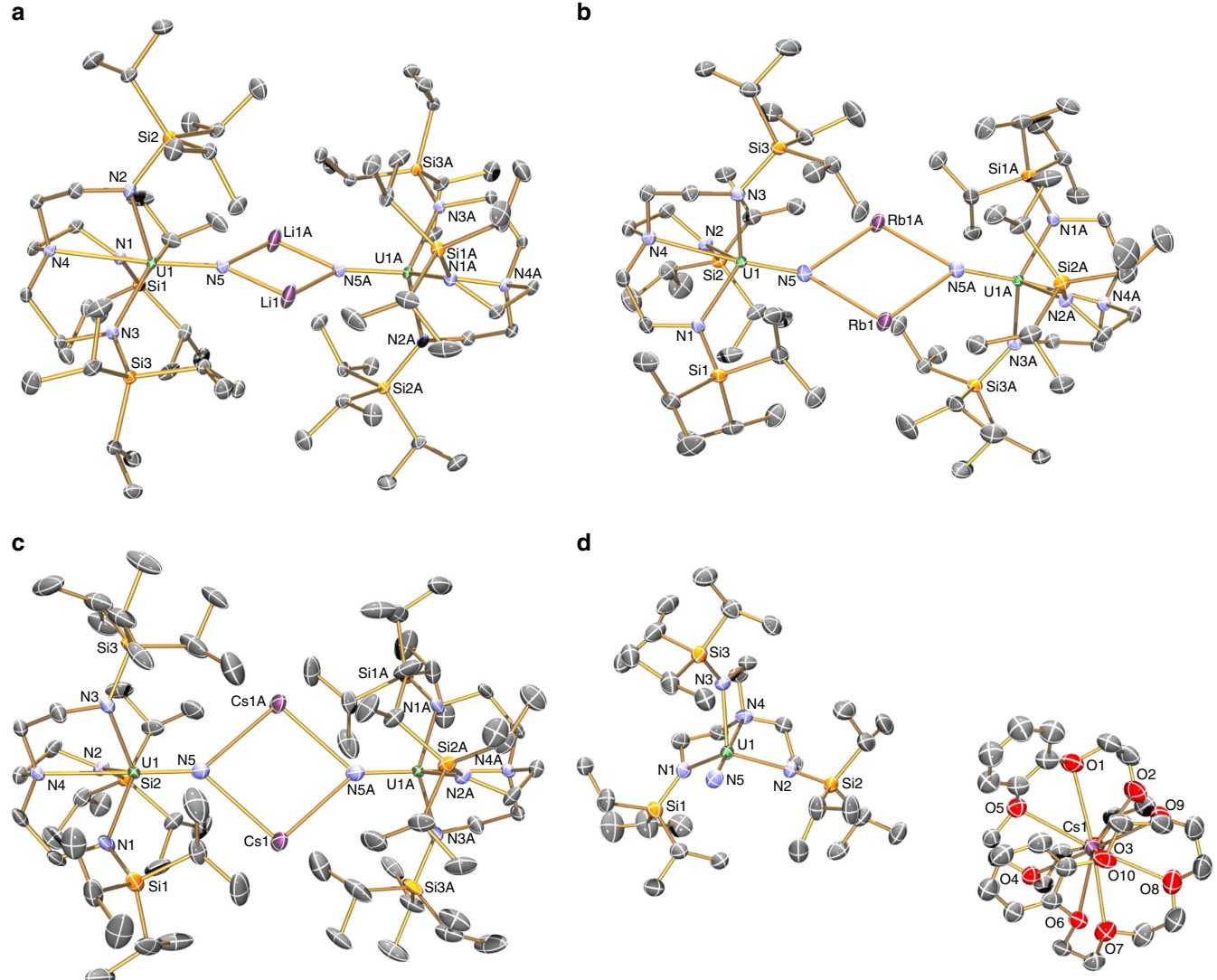

**Figure 2 | Crystal structures of 3M and 4M.** Molecular structures with displacement ellipsoids at 40%, and minor disorder, hydrogen atoms, counter ions and lattice solvent omitted, with selective labelling of: (**a**) **3Li**, (**b**) **3Rb**, (**c**) **3Cs** and (**d**) **4Cs**. The structures of **3Na**, **4Na** and **3K** have been reported previously[24,27], but the latter was previously prepared by $KC_8$ reduction of **6** rather than **7**. We report here a new polymorph of **3K** (Supplementary Methods). The anion components of **4Na–4Cs** are very similar and differ mainly in only the nature of the alkali metal bis(crown) cation component.

CF + SOC parameterization requires new terms appropriate to the $C_3$ point group. In addition to the $q = 0$ terms, $|q| = 3$ and 6 terms are allowed: these mix states with $\Delta l_z = 3$ and 6, respectively, hence mixing $l_z = \pm 3$ with 0, $l_z = \pm 2$ with $\pm 1$ and $l_z = +3$ with $-3$ (Fig. 4). After SOC, this can also be described as $\Delta j_z = 3$ or 6 mixing. Considering the threefold CF as a perturbation to the axial field, its effect will be greatest for states that are closest in energy. Hence, direct mixing between $l_z = +3$ and $-3$ (breaking their degeneracy) under $|q| = 6$ will be most significant. Any mixing induced by $|q| = 3$ would not induce any further loss of degeneracy and hence their effects can be parameterized with the $q = 0$ terms. Therefore, to minimize the number of parameters, we have used a truncated Hamiltonian (neglecting $|q| = 3$ terms), namely, the $D_3$ Hamiltonian $\hat{H} = B_2'^0\hat{O}_2^0 + B_4'^0\hat{O}_4^0 + B_6'^0\hat{O}_6^0 + B_6'^6\hat{O}_6^6 + \lambda\hat{l}\cdot\hat{s}$, where we have also chosen our coordinate system such that $B_6'^{-6} = 0$ (ref. 56). Fitting to the *ab initio* orbital energies ($\lambda = 0$) gives the CF parameters (Supplementary Table 6), and we find $B_6'^6$ to be $\sim 1\%$ of $B_2'^0$. Fixing these parameters and varying $\lambda$ to fit to the *ab initio* SOC-doublet energies gives $\lambda = 2{,}022\,\mathrm{cm}^{-1}$.

However, this gives $g_z = 4.27$ for the ground doublet, significantly larger than the *ab initio* value of $g_z = 3.80$. Attempts to remedy this by inclusion of $|q| = 3$ terms requires unfeasibly large parameters (due to the large energy gap to the $l_z = 0$ state with which the predominantly $|l_z, s_z, j_z\rangle = |\pm 3, \mp 1/2, \pm 5/2\rangle$ ground doublet can mix via $\hat{O}_k^3$). Therefore, it is likely that the inconsistency lies in the basis states, which are pure $5f$ functions, while the orbitals we are trying to describe are MOs with appreciable ligand character. This can be treated with an orbital reduction parameter $0 \leq \kappa \leq 1$ (note this is the Greek letter $\kappa$, not to be confused with the rank of the CF operators, k)[57]. Thus, our final parametric Hamiltonian, including the Zeeman term, becomes:

$$\hat{H} = \kappa^2 B_2'^0\hat{O}_2^0 + \kappa^4 B_4'^0\hat{O}_4^0 + \kappa^6\left(B_6'^0\hat{O}_6^0 + B_6'^6\hat{O}_6^6\right) + \kappa\lambda\hat{l}\cdot\hat{s} + \mu_B\left(\kappa\hat{l} + 2\hat{s}\right)\cdot B \quad (1)$$

Fixing the off-axial CF term ($\kappa^6 B_6'^6$) to the value in Supplementary Table 6, we find a reasonable $\kappa = 0.92$ reproduces the *ab initio* ground state $g_z$ (Supplementary Table 7).

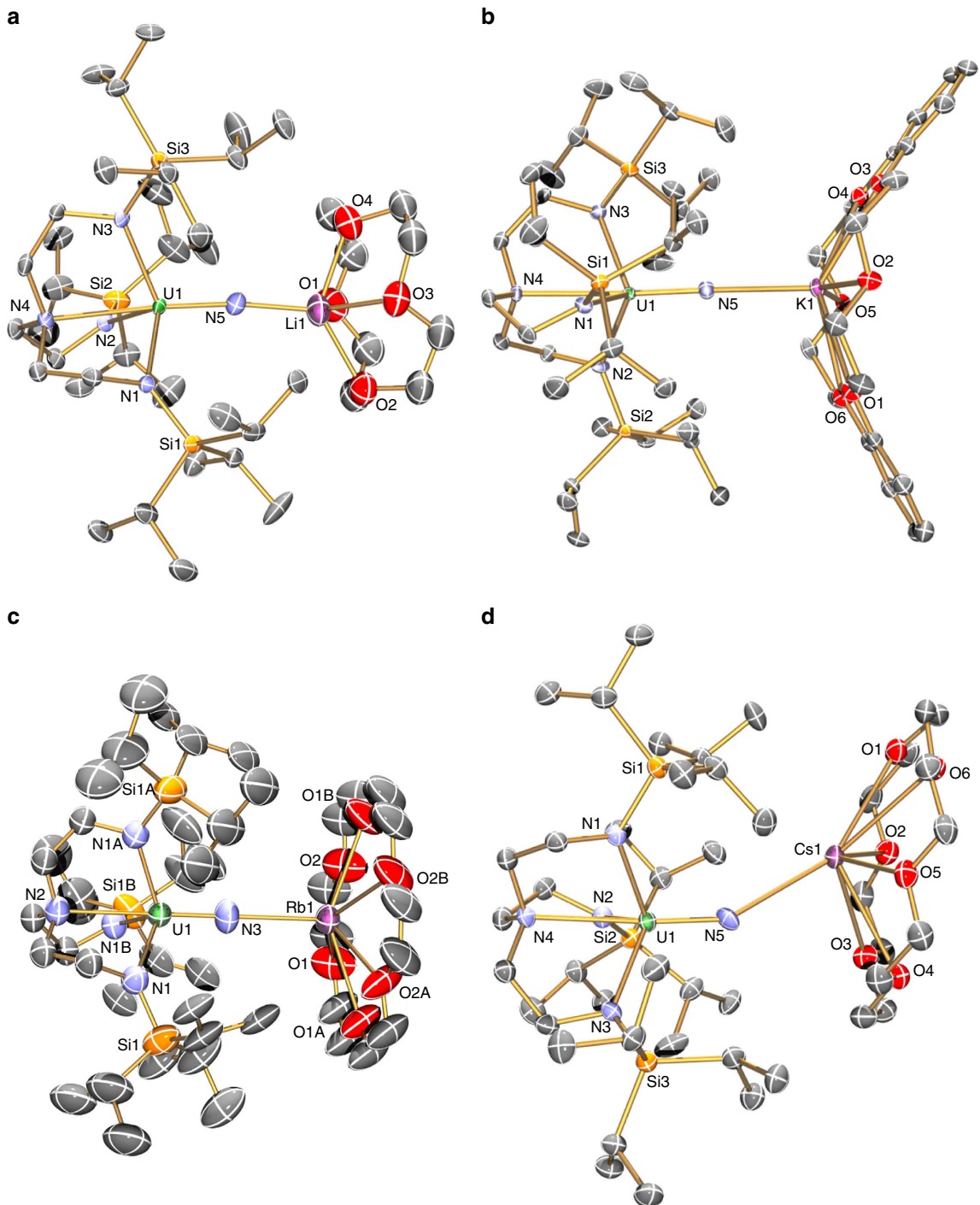

**Figure 3 | Crystal structures of 5M.** Molecular structures with displacement ellipsoids at 40%, and minor disorder, hydrogen atoms, counter ions and lattice solvent omitted, with selective labelling of: (**a**) **5Li**, (**b**) **5K**, (**c**) **5Rb** and (**d**) **5Cs**. The structure of **5Na** has been reported previously[25].

*Ab initio* calculations on model complexes deviating from $C_3$, and building up to the full species, namely, non-symmetrized $[U(NH_3)(NH_2)_3(N)]^-$, $[U(Tren^H)(N)]^-$ and, finally, $[U(Tren^{TIPS})(N)]^-$, show that lower symmetry perturbations introduce only small splittings within the $l_z = \pm 2$ and $\pm 1$ pairs compared with the dominant axial and $C_3$ fields (Fig. 5c–e). In fact, there is little difference in the *ab initio* results across **3M–5M** save for the energies and character of the third and fourth excited doublets (Supplementary Tables 8, 9 and 10; averaged energies in Table 1 and Fig. 5f). These latter states, which are predominantly $j_z \approx \pm 5/2$ and $\pm 7/2$, are close in energy and directly mixed via the $|q| = 6$ terms, thus are sensitive to small structural changes.

**EPR and UV/Vis/NIR Spectroscopic Studies**. Solid-state EPR spectra are dominated by an absorption-like feature at $g \sim 3.74(9)$ for all compounds except **3Li** and **3Na**. This indicates an axial $g_z$ feature with small, but non-zero, $g_{x,y}$ (Fig. 6a; Supplementary Figs 3–14; Supplementary Table 11). We do not observe $g_{x,y}$ within the magnetic field range, implying $g_{x,y} < \sim 0.5$. This is in remarkable agreement with the *ab initio* calculations of $g_z \sim 3.84$ and $g_{x,y} \sim 0.3$ for the $j_z \approx \pm 5/2$ doublet. For dinuclear **3Li** and **3Na**, additional fine-structure is observed, originating from exchange interactions (see below); this is most clear at Q-band, where the $g_z$ feature is split into a doublet (Fig. 6e; Supplementary Figs 15 and 16).

UV/Vis/NIR absorption spectra of **4M** and **5M** in toluene are all similar (Fig. 6b; Supplementary Fig. 17; **3M** are insoluble in non-polar solvents). Spectra are dominated by ligand-to-metal charge transfer bands that tail in from the ultraviolet region to $\sim 12,000\,cm^{-1}$. Five $f$–$f$ transitions are observed at $\sim 4,700$, $\sim 6,000$, $\sim 6,900$ and $\sim 8,750\,cm^{-1}$ ($\varepsilon \sim 20$–$25\,M^{-1}\,cm^{-1}$) with a shoulder on the ligand-to-metal charge transfer at $\sim 18,000\,cm^{-1}$ ($\sim 250\,M^{-1}\,cm^{-1}$), in excellent agreement with the *ab initio* energies (Table 1; Supplementary Table 12).

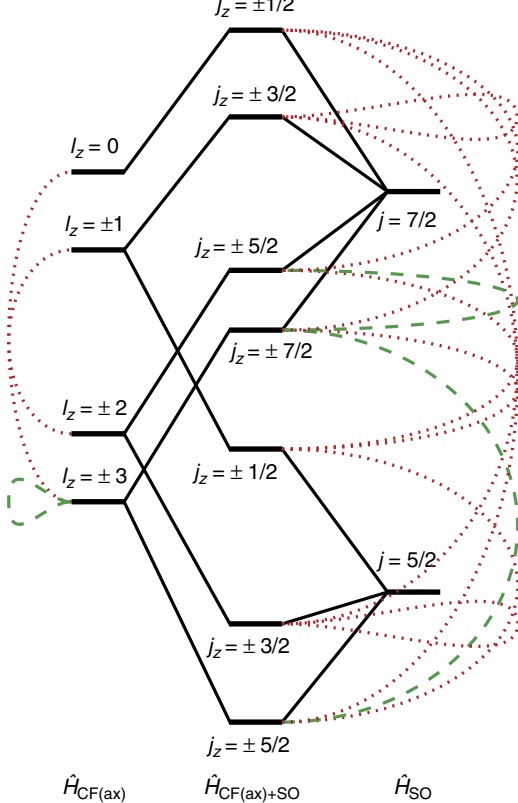

**Figure 4 | Crystal field and SOC for f[1].** Simplified diagram of states arising from the $\hat{H}_{CF(ax)}$, $\hat{H}_{SO}$ and combined $\hat{H}_{CF(ax)+SO}$ Hamiltonians. Green dashed lines indicate states that have $\Delta l_z$ or $\Delta j_z = 6$ and red dotted lines indicate states that have $\Delta l_z$ or $\Delta j_z = 3$, corresponding to states that are mixed upon lowering to $C_3$ symmetry. Note that this is a simplified picture that does not depict the mixing between $\Delta j_z = 0$ terms, for example, the lowest $j_z = \pm 5/2$ state mixes with the upper $j_z = \pm 5/2$ state and therefore derives partially from the $j = 7/2$ SOC state as well as the $l_z = \pm 2$ CF state.

**Magnetic studies.** For all compounds, the room temperature $\chi_M T$ values per uranium ($\chi_M$ is the molar magnetic susceptibility) lie in the narrow range 0.38–0.49 cm$^3$ K mol$^{-1}$: these are towards the lower end of the range associated with uranium(V) (0.25–1.3 cm$^3$ K mol$^{-1}$)[58] due to the strong CF from the nitride. $\chi_M T$ decreases slowly on cooling in the high-temperature range (Fig. 7a–c). For **4M**, $\chi_M T$ decreases more rapidly below $\sim 50\,K$ reaching 0.1–0.2 cm$^3$ K mol$^{-1}$ at 2 K. For **3M** and **5M**, $\chi_M T$ decreases more slowly, reaching 0.3–0.4 cm$^3$ K mol$^{-1}$ (**3K** and **5Na** exhibit little temperature dependence). The only outlier in this trend is **5K**, which seems to behave more like the **4M** series.

A more accurate measurement of $\chi_M T$ is from the in-phase component of the ac susceptibility ($\chi_M'T$; Supplementary Figs 18, 19 and 20), because the differential susceptibility ($\partial M/\partial H$) is measured directly by the small oscillating field ($\sim 1.5\,G$). For **3K**, **3Rb**, **3Cs** and **5M** (excluding **5K**), measured in static field $H = 0$ or 0.1 T, the frequency-independent part of $\chi_M'T(T)$ plateaus at low temperatures (at 0.27–0.37 cm$^3$ K mol$^{-1}$ per uranium; Supplementary Figs 18 and 20), implying that the ground doublet is isolated under these conditions. In contrast, $\chi_M'T(T)$ fails to plateau for **4M** and **5K** (Supplementary Figs 19 and 20), implying that the first excited doublet is closer in energy. Extrapolating $\chi_M'T(T)$ to 0 K for **4M** and **5K** gives significantly lower values (0.1–0.2 cm$^3$ K mol$^{-1}$), implying that the lowest doublet is less magnetic. Low temperature magnetization ($M$) data mirror these trends (Fig. 7d–f): **3M** and **5M** (excluding **5K**) reach 0.6–0.8 $\mu_B$ mol$^{-1}$ at 1.8 K and 7 T, while **4M** and **5K** reach lower values of 0.3–0.6 $\mu_B$ mol$^{-1}$.

For all complexes, except **3Li**, $\chi_M'T(T)$ decreases with increasing a.c. frequency below 10 K, reaching zero (0.1 T static field). This is accompanied by the appearance of peaks in the out-of-phase susceptibility ($\chi_M''$) as a function of frequency and temperature (Fig. 8; Supplementary Fig. 21). This shows slow relaxation of the magnetization, hence these materials are behaving as SMMs. Only weak frequency dependence is observed for nil static field. Hysteresis is not observed in $M(H)$ experiments.

The exchange-coupled dimers **3Li** and **3Na** behave differently from the other **3M** complexes: $\chi_M'T(T)$ fails to plateau at low temperature (Supplementary Fig. 18) and $M(H)$ exhibits waist-restricted hysteresis that collapses below $\sim 0.5\,T$ (Fig. 7a; Supplementary Fig. 16).

## Discussion

The few methods of preparing isolable uranium nitrides[30–41] contrast to the plethora of methods in the transition metal series[46,47], and for uranium most involve redox activity of azide. In this work, we have found new and reliable methods to a wide family of terminal uranium nitrides based on the U(Tren$^{TIPS}$)

**Table 1 | *Ab initio* and optical results.**

| SO state | Ab initio | | | Experimental | |
| --- | --- | --- | --- | --- | --- |
| | Energy (cm$^{-1}$) | $g_z$ | $g_{x,y}$ | Energy (cm$^{-1}$)* | $g_z$† |
| $j_z \approx \pm 5/2$ | 0(100) | 3.84(8) | 0.3(2) | — | 3.74(9) |
| $j_z \approx \pm 3/2$ | 600(100) | 2.4(3) | 0.3(2) | — | — |
| $j_z \approx \pm 1/2$ | 4,460(70) | 0.25(5) | 1.7(1) | 4,700(100) | — |
| — | 6,700(100) | — | — | 6,060(20) | — |
| — | 7,500(200) | — | — | 6,900(200) | — |
| $j_z \approx \pm 3/2$ | 9,400(100) | 3.7(1) | 0.1(1) | 8,900(500) | — |
| $j_z \approx \pm 1/2$ | 15,800(600) | 1.84(2) | 3.5(1) | 18,000(1,000) | — |

EPR, electron paramagnetic resonance, SOC, spin–orbit coupling; UV/Vis, ultraviolet/visible.
Comparison between average *ab initio* predictions and average experimental observations of the SOC states for the series **3M**, **4M** and **5M**. S.d.'s are given in brackets.
*From UV/Vis.
†From EPR.

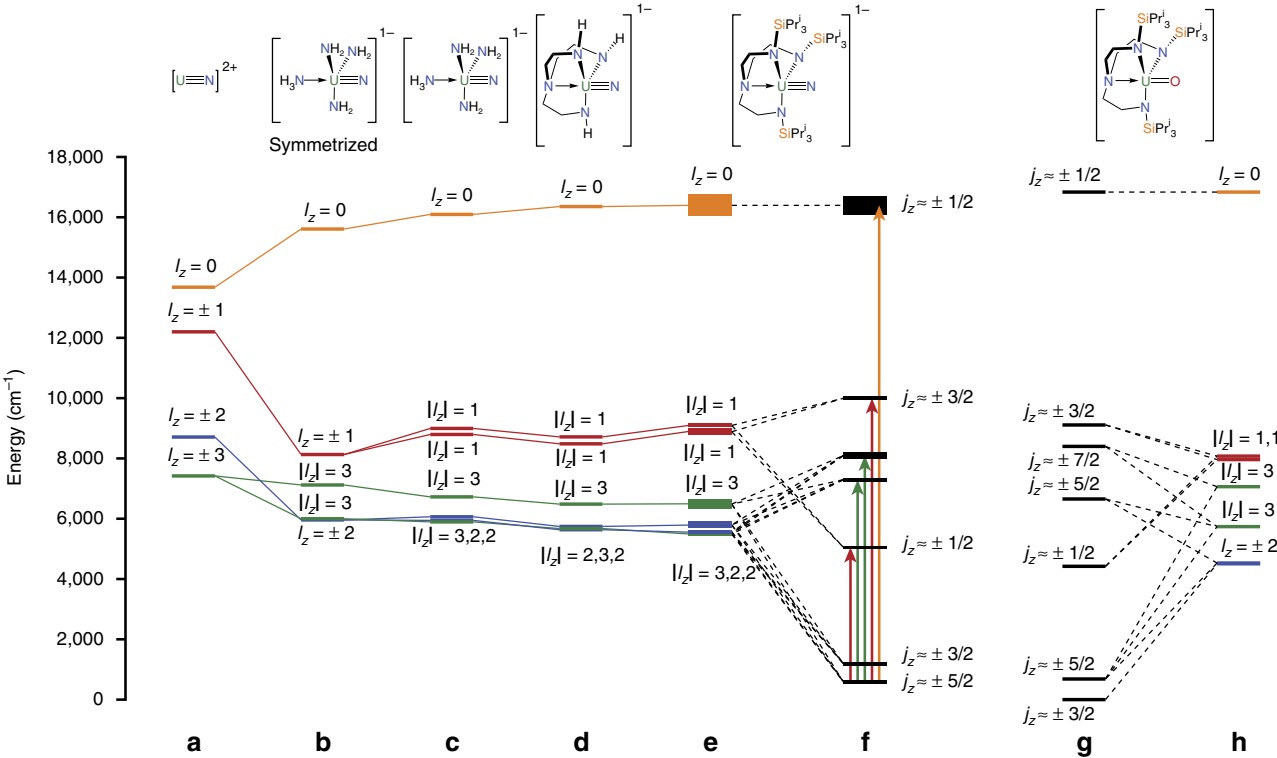

**Figure 5 | Level diagrams for nitrides and oxide.** 5$f$ orbital energy diagram for model uranium(V) complexes from *ab initio* calculations; (**a**) [UN]$^{2+}$, (**b**) symmetrized [U(NH$_3$)(NH$_2$)$_3$(N)]$^-$, (**c**) low symmetry [U(NH$_3$)(NH$_2$)$_3$(N)]$^-$, (**d**) [U(Tren$^H$)(N)]$^-$, (**e**) average **3M**, **4M** and **5M** of [U(Tren$^{TIPS}$)(N)]$^-$, (**f**) average **3M**, **4M** and **5M** of [U(Tren$^{TIPS}$)(N)]$^-$ with SOC, (**g**) comparative terminal mono(oxo) complex **1** [U(Tren$^{TIPS}$)(O)] with SOC, and (**h**) comparative terminal mono(oxo) complex **1** [U(Tren$^{TIPS}$)(O)]. The thickness of the levels for **e**,**f** indicates the s.d. around the average positions across the series **3M**, **4M** and **5M**. Third and fourth excited SO-coupled states for **f** are not given approximate $j_z$ labels, as the composition of these states varies greatly across the series **3M**, **4M** and **5M**. The arrows refer to optical transitions within the $j_z$ electronic manifold from ground to excited states and the colour references the orbital $l_z$ parentage of the given transition.

framework (Supplementary Note 2), and it is our hope that this could be applicable to other ligand systems. We note that the reactivity described here is analogous to that reported recently for a titanium nitride[59], since uranium is behaving like an early *d*-block ion in this context.

Given the simplicity of our *ab initio* approach, there is remarkable agreement between calculated and the experimental energies from UV/Vis/NIR (deviations of up to $\sim 10^3$ cm$^{-1}$, or only 2.9 kcal mol$^{-1}$). The transition to the first excited doublet is outside our spectral range, but all other states are observed. Interestingly, the *f–f* transition linewidths follow a consistent pattern (from low to high energy) of broad, sharp, sharp, broad and very broad. These transitions are to excited states derived primarily from $|l_z| = 1$, $|l_z| = 2$ or 3, $|l_z| = 2$ or 3, $|l_z| = 1$ and $|l_z| = 0$, respectively (Fig. 5f). The broad transitions are to states with $|l_z| = 1$ and 0, deriving from orbitals that are formally anti-bonding with respect to the nitride, while the sharp transitions involve non-bonding $|l_z| = 2$ or 3 states. The former are expected to be much more susceptible to U–N vibrational broadening (and more so for $|l_z| = 0$ than 1) than the latter. Hence, the assignments are consistent with the orbital parentage of the SO-coupled states. If this is characteristic for uranium(V) more widely, it could serve as a tool to directly evidence the nature of SO-coupled states, although it may only apply in cases with dominant axial CFs.

Importantly, the gross electronic structure is clear—there are two low-lying doublets ($j_z \approx \pm 5/2$ and $\pm 3/2$), within a few hundred cm$^{-1}$ of each other, well separated from all others by thousands of cm$^{-1}$. In fact, the calculated separation of the two

lowest doublets is of the same order as the errors in the calculations. Hence, it is feasible that their actual ordering could be reversed and dependent on small structural differences. (Note there is no significant difference in the *ab initio* results upon enlargement of the active space and basis set, or correcting for dynamic correlation; Supplementary Table 13.) However, these two doublets will dominate the magnetic properties. EPR gives $g_z \approx 3.74$ (9) with an upper limit for $g_{x,y} < 0.5$, in agreement with *ab initio* values for the $j_z \approx \pm 5/2$ doublet of 3.84 (8). The $j_z \approx \pm 3/2$ doublet is calculated to have $g_z \approx 2.4$ (3). Hence, the EPR spectra must arise from the $j_z \approx \pm 5/2$ doublet in each case, and we initially assumed that this was the ground state. However, this is not compatible with the magnetic data in all cases.

The calculated magnetization for an isolated $j_z \approx \pm 5/2$ doublet with $g_z = 3.74$, $g_{x,y} = 0$ is 0.90 $\mu_B$ mol$^{-1}$ (for 1.8 K and 7 T; Fig. 7), with a $\chi_M T$ value of 0.44 cm$^3$ K mol$^{-1}$. The equivalent values for an isolated $j_z \approx \pm 3/2$ doublet ($g_z = 2.40$, $g_{x,y} = 0$) are 0.55 $\mu_B$ mol$^{-1}$ and 0.18 cm$^3$ K mol$^{-1}$. The latter are consistent with experimental data for **4M** and **5K** ($M = 0.33$–0.53 $\mu_B$ mol$^{-1}$; extrapolated 0 K $\chi_M'T = 0.12$–0.22 cm$^3$ K mol$^{-1}$), particularly allowing for uncertainty in $g_z$. Hence, it appears for **4M** and **5K** that $j_z \approx \pm 3/2$ is the ground state with $\pm 5/2$ as a low-lying excited state, consistent with the significant temperature dependence in $\chi_M T$. The $j_z \approx \pm 5/2$ state must be low enough that it is still sufficiently populated at $< 10$ K to give the observed EPR spectrum. The experimental data for **3K**, **3Rb**, **3Cs** and **5M** (excluding **5K**) are significantly higher ($M = 0.6$–0.75 $\mu_B$ mol$^{-1}$; extrapolated 0 K $\chi_M'T = 0.26$–0.37 cm$^3$ K mol$^{-1}$ per uranium). This implies that $j_z \approx \pm 5/2$ is significantly lower in energy.

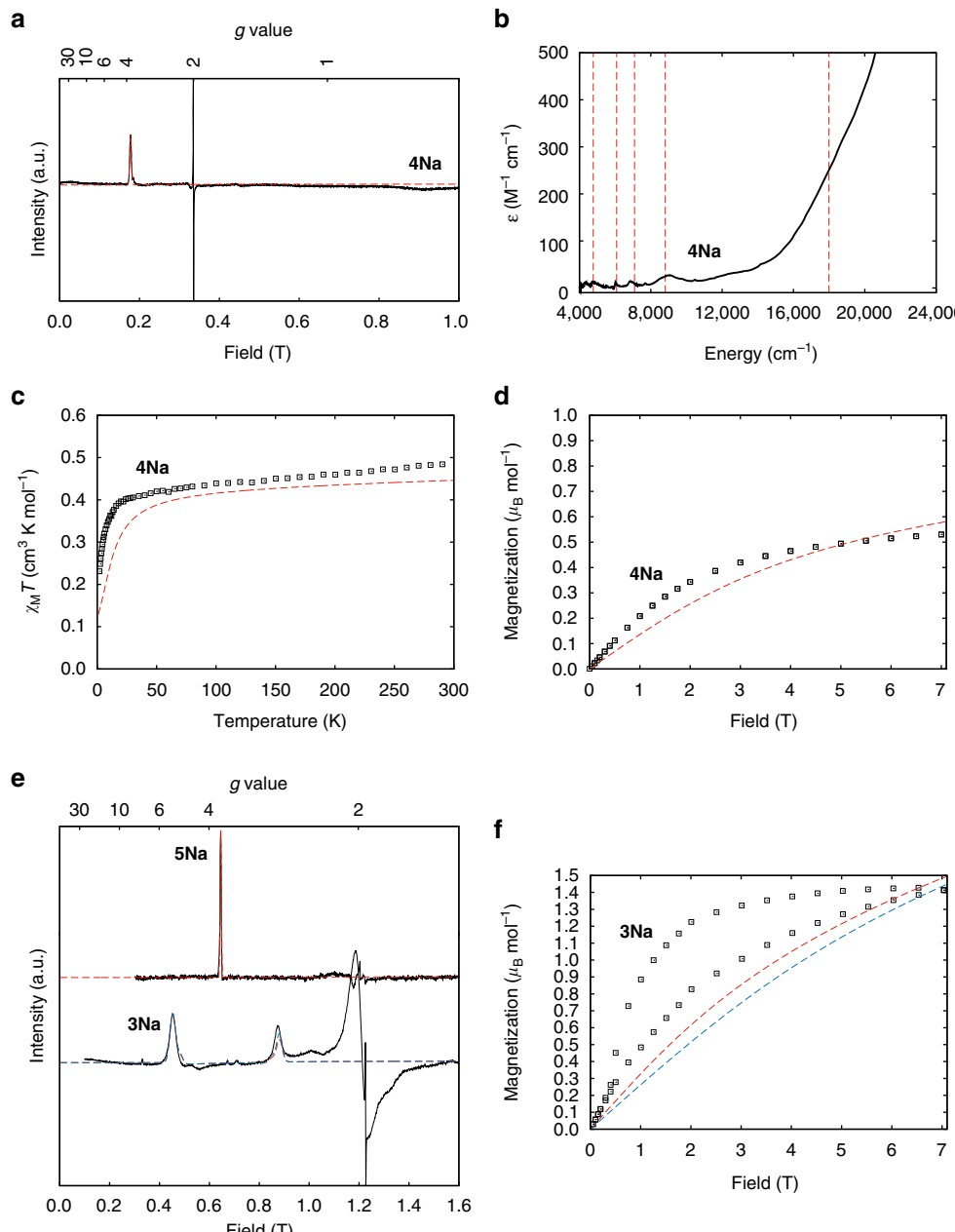

**Figure 6 | EPR and magnetic data for 3Na–5Na.** (**a**) X-band EPR spectrum at 5 K, (**b**) NIR/Vis/UV in toluene at room temperature, (**c**) $\chi_M T$ in 0.1 T field, (**d**) and magnetization at 1.8 K for **4Na**: experimental (black curves and points) and fitted (dashed curves). (**e**) Q-band EPR spectra of **5Na** and **3Na** at 5 and 7 K, respectively: fits use Equation 1 for **5Na** (red) and Equation 2 for **3Na** with $j_{zz} = +0.7678$ cm$^{-1}$ (red) and $j_{zz} = -0.7678$ cm$^{-1}$ (blue). (**f**) Magnetization (per dimer) for **3Na** at 1.8 K: the slow magnetization dynamics as evidenced by the magnetic hysteresis has not been simulated; the equilibrium magnetization is shown and differs depending on the sign of the exchange interaction.

Indeed, for **3K** and **5Na**, the data approach the calculated values for $j_z \approx \pm 5/2$ and, along with the lack of temperature dependence in $\chi_M T$, suggests that this is the ground state.

To estimate the separation of the two doublets for each complex, we attempted to fit $\chi_M T(T)$ while simultaneously fitting all other experimental data (EPR, optical and magnetization) to the CF + SOC Hamiltonian (Equation 1; using PHI software;[60] see Methods). To avoid over-parameterization, $B_6'^6$ was fixed at 1% of $B_2'^0$ as from *ab initio* calculations (Supplementary Tables 6 and 7). For **3M**, where we do not have UV/Vis/NIR data, and for the highest-energy level of **4Na** that is not experimentally resolved, average transition energies from the other species were employed (Table 1). The fits are good considering the

breadth of data (Fig. 6; Supplementary Figs 3–15), giving the parameters in Table 2.

By modelling all the experimental data with a single model, we can discuss features of the experimental electronic structure. For the 5*f*-orbital energies, for all complexes the barycentre of the $l_z = \pm 3$ orbitals is 800(200) cm$^{-1}$ above the $\pm 2$ pair, followed by $l_z = \pm 1$ and 0 at 3,500(200) and 13,800(400) cm$^{-1}$, respectively. The splitting of the $l_z = \pm 3$ pair is 1,290(70) cm$^{-1}$ (except for **4Rb** where it is 1,630 cm$^{-1}$), giving a gap between the lowest lying orbitals of only 200–300 cm$^{-1}$, hence the subtlety of the electronic structure of these compounds. With SOC, the fit parameters give the ground state as $j_z \approx \pm 5/2$ for **3K** and **5Na** with the $\pm 3/2$ state lying at 300–400 cm$^{-1}$;

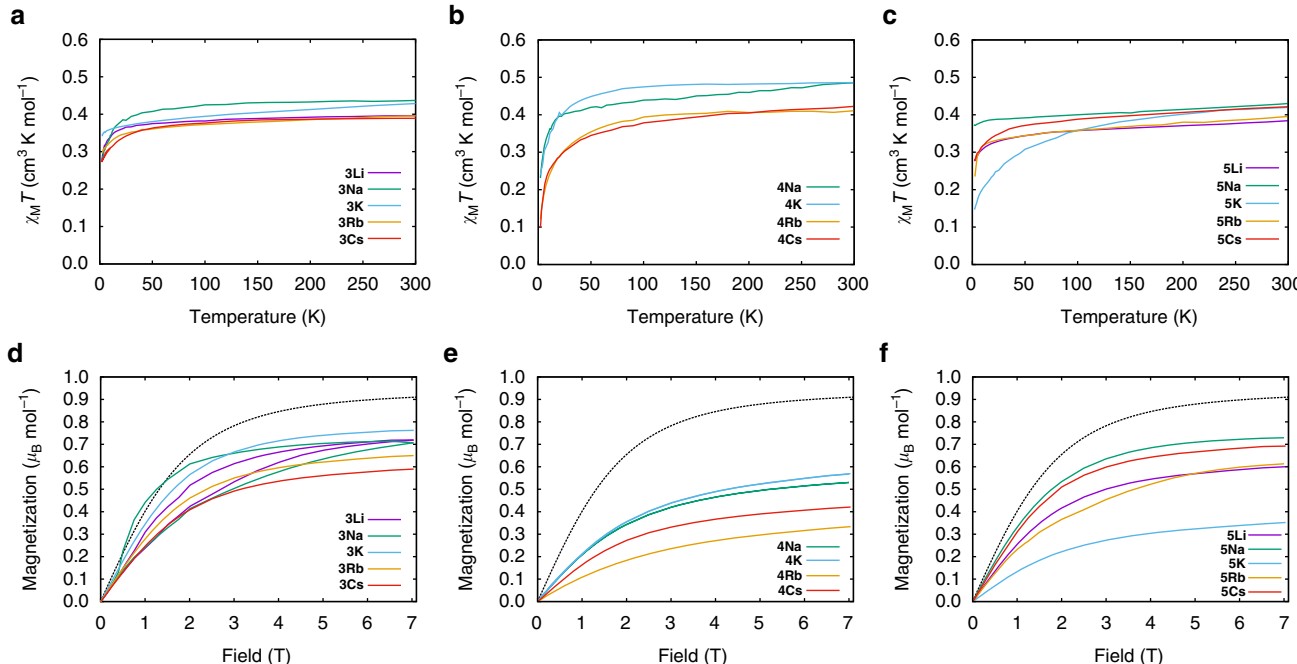

**Figure 7 | Magnetic data.** Experimental $\chi_M T$ curves in a 0.1 T d.c. field; (**a**) DCIP series **3M**, (**b**) SIP series **4M** and (**c**) CIP series **5M**, and experimental magnetization curves at 1.8 K; (**d**) DCIP series **3M**, (**e**) SIP series **4M** and (**f**) CIP series **5M**. Black dashed curve gives expected magnetization for $j_z \approx \pm 5/2$ state with $g_{x,y} = 0$ and $g_z = 3.74$. All data are shown per uranium.

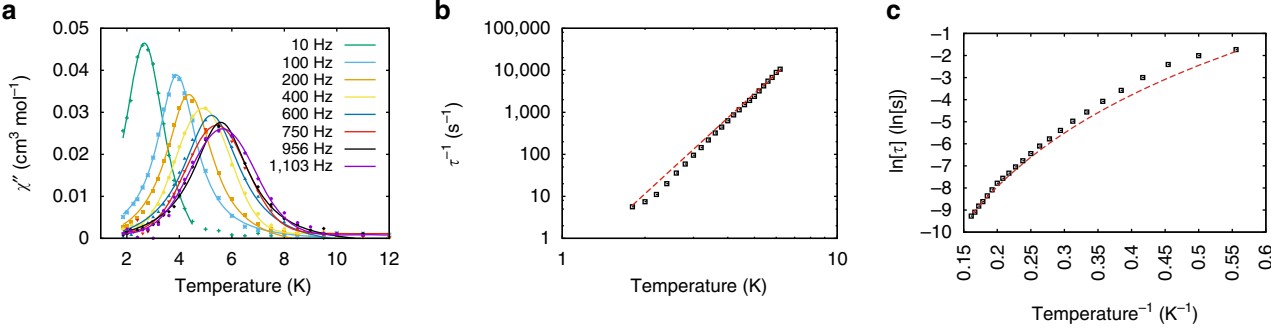

**Figure 8 | AC magnetic data for 5Na.** Experimental (points) and fitted (coloured curves) out-of-phase susceptibility $\chi''$ (**a**) and magnetization relaxation times (**b**,**c**) for **5Na** as a representative example. AC data measured under a 0.1 T field.

however, this gap is poorly defined as we have no direct experimental handle. For all other complexes, the fits give the ground state as $j_z \approx \pm 3/2$ with the $\pm 5/2$ state at 7–46 cm$^{-1}$, where this gap is now better defined by the temperature dependence of $\chi_M T$. Higher excited states are as in Table 1. Compared with the well-characterized octahedral uranium(V) hexahalides[11] and alkyl (R), alkoxide (OR), amide (NR$_2$) and ketimide (NCR$_2$) [UX$_6$]$^-$ complexes[8], where the 5$f$-orbitals span 11,361 (X = F), 6,307 (Cl), 5,310 (Br), 6,800 (R), 8,500 (OR), 7,400 (NR$_2$) and 6,900 cm$^{-1}$ (NCR$_2$), the total breadth of the 5$f$-orbitals here spans a much larger 13,800 (400) cm$^{-1}$ consistent with the strong axial CF of the nitride. The effective SOC constants are practically identical, where we find $\kappa\lambda = 1,900(20)$ cm$^{-1}$ compared with 1,910(70) cm$^{-1}$ for [UX$_6$]$^-$ (X = F, Cl, Br, R, OR, NR$_2$ and NCR$_2$).

Our model gives $1.9 < g_z < 2.2$ with $g_{x,y} = 0$ for the $j_z \approx \pm 3/2$ state, and $3.7 < g_z < 4.0$ with $0.14 < g_{x,y} < 0.18$ for the $j_z \approx \pm 5/2$ state. With these values, the $j_z \approx \pm 5/2$ state is EPR active ($g_{x,y} \neq 0$), while the $\pm 3/2$ state is EPR silent ($g_{x,y} = 0$). This is consistent with

the observation that we do not observe EPR transitions corresponding to the $j_z \approx \pm 3/2$ doublet regardless of the ground state. This is because in $C_3$ symmetry the $j_z \approx \pm 5/2$ state mixes with $\pm 7/2$ under the $|q| = 6$ terms (Equation 1; Fig. 4), hence introducing a $\Delta j_z = \pm 1$ component (the EPR selection rule). In contrast, $j_z \approx \pm 3/2$ does not mix with any other state under $|q| = 6$; even $|q| = 3$ terms could only mix it with the other $j_z \approx \pm 3/2$ state. No additional mixing is induced by SOC and there is no $\Delta j_z = \pm 1$ component. Although *ab initio* calculations give a small $g_{x,y} \neq 0$ for this state, which must be due to low symmetry effects, our experiments are consistent with $C_3$ symmetry.

We previously reported magnetic data for the isoelectronic mono(oxo) complex **1** (ref. 14). Compound **1** has a low temperature plateau in $\chi_M' T$ of 0.18 cm$^3$ K mol$^{-1}$, a saturation magnetization of 0.52 $\mu_B$ mol$^{-1}$ at 1.8 K, and is EPR silent. These data agree with a well-isolated $j_z \approx \pm 3/2$ ground state, in contrast to the $j_z \approx \pm 5/2$ or near-degenerate $\pm 5/2$, $\pm 3/2$ states of the nitrides. This can be rationalized with simple CF arguments: (i) **1** has a formal U $=$ O double bond so gives a weaker axial CF

**Table 2 | Model parameters.**

| Parameter | 3Na | 3K | 3Rb | 3Cs | 4Na | 4K | 4Rb | 4Cs | 5Li | 5Na | 5K | 5Rb | 5Cs |
|---|---|---|---|---|---|---|---|---|---|---|---|---|---|
| $B'^0_2$ (cm$^{-1}$) | −353.7 | −341.7 | −331.8 | −332.1 | −322.6 | −322.8 | −274.9 | −325.8 | −340.0 | −331.7 | −324.5 | −319.0 | −321.7 |
| $B'^0_4$ (cm$^{-1}$) | 17.36 | 14.89 | 15.75 | 15.78 | 15.11 | 15.36 | 11.54 | 15.60 | 16.11 | 14.14 | 15.42 | 15.03 | 15.34 |
| $B'^0_6$ (cm$^{-1}$) | −2.138 | −2.009 | −1.910 | −1.916 | −1.810 | −1.835 | −1.182 | −1.874 | −2.051 | −1.761 | −1.781 | −1.699 | −1.753 |
| $\|B'^6_6\|$ (cm$^{-1}$)* | 3.537 | 3.417 | 3.318 | 3.321 | 3.226 | 3.228 | 2.749 | 3.258 | 3.400 | 3.317 | 3.245 | 3.190 | 3.217 |
| $\lambda$ (cm$^{-1}$) | 2121 | 2078 | 2112 | 2111 | 2110 | 2112 | 1956 | 2118 | 2107 | 2054 | 2115 | 2117 | 2121 |
| $\kappa$ | 0.8845 | 0.9004 | 0.9027 | 0.9025 | 0.9112 | 0.9011 | 0.9681 | 0.8994 | 0.9072 | 0.9288 | 0.9030 | 0.9032 | 0.8968 |
| $j_{zz}$ (cm$^{-1}$) | ± 0.7678 | — | — | — | — | + 0.01297 | — | — | — | — | — | — | — |
| X-band linewidth (GHz)† | 1.4 | 0.3648 | 0.5816 | 0.6004 | 0.2576 | 0.2328 | 0.3161 | 0.3921 | 0.3720 | 0.2636 | 0.3878 | 0.2940 | 0.4750 |
| Q-band linewidth (GHz)† | 1.353 | 0.4176 | 0.7828 | 0.6128 | 0.2743 | 0.2111 | 0.8241 | 0.6589 | 0.4467 | 0.2965 | 0.4724 | 0.3853 | 0.2331 |

CF, crystal field; SO, spin orbital.
Parameters derived from fitting experimental data with CF and SO model (Equation 1).
*Fixed at 1% of $\|B'^0_2\|$.
†Fixed at the values derived from the effective $s = 1/2$ model (Supplementary Table 15), excepting **3Na** and **4K**.

than a formal U≡N triple bond; (ii) the equatorial CF due to the Tren$^{TIPS}$ ligand is unchanged; and (iii) the weaker axial CF of the oxo stabilizes the $l_z = \pm 2$ relative to $\pm 3$ orbitals, therefore stabilizing the $j_z \approx \pm 3/2$ doublet relative to $\pm 5/2$ after SOC. *Ab initio* calculations on **1** (Fig. 5g,h; Supplementary Table 14) give the ground state as $j_z \approx \pm 3/2$ with the $\pm 5/2$ first excited state at 683 cm$^{-1}$; that is, compared with the *ab initio* results on the nitrides, the $j_z \approx \pm 3/2$ doublet has been stabilized relative to $\pm 5/2$ by $\sim 1,500$ cm$^{-1}$. Although there is some variation among **3M–5M**, these differences are subtle compared with the difference between U = O and U≡N complexes (Fig. 5).

Q-band EPR spectra of the DCIP complexes **3Li** and **3Na** show a pair of transitions centred on $g_z \approx 3.7$ (Fig. 6e) due to exchange interactions within the {U≡N-(M)$_2$-N≡U} fragments. While U$^V$···U$^V$ and even Np···Np exchange have been determined from magnetic data[9,61–65], as far as we are aware this is the first example of direct spectroscopic measurement of such an interaction. Treating the two uranium(V) ions as effective $s = 1/2$ species with $g_{x,y} = 0$, and given the near co-linearity of the U≡N vectors, the interaction can be described by the Ising Hamiltonian $\hat{H}_{int} = -2j_{zz}\hat{s}_{z_1}\hat{s}_{z_2}$. The resulting eigenfunctions are a magnetic and a non-magnetic doublet ($|\uparrow\uparrow\rangle, |\downarrow\downarrow\rangle$ and $|\downarrow\uparrow\rangle, |\uparrow\downarrow\rangle$, respectively), separated by $|j_{zz}|$. For EPR microwave energy $h\nu \gg |j_{zz}|$, two transitions arise separated by $|j_{zz}|/g_z\mu_B$ in field units. Modelling the spectrum gives $j_{zz} = -0.37$ cm$^{-1}$ for **3Na** (the relative intensity of the two peaks gives the sign; Supplementary Table 15). This value is confirmed by a near zero-field resonance at X-band where $h\nu \approx |j_{zz}|$ (Supplementary Fig. 15). Spectra of **3Li** are more complex (Supplementary Fig. 16) but the splitting between the dominant peaks is similar. This $j_{zz}$ is much larger than the dipolar interaction calculated from $g_z$ and the U···U geometry ($\sim 0.005$ cm$^{-1}$; Supplementary Table 16), and thus indicates a super-exchange component through the NM$_2$N bridge. Although EPR spectra of **3K–3Cs** only show a single $g_z \approx 3.7$ transition, the linewidth is significantly broader than the monometallic complexes; modelling this as due to U$^V$···U$^V$ interactions gives values close to the calculated dipolar values (Supplementary Table 16). Hence, there is a weak but spectroscopically significant exchange coupling in **3Li** and **3Na**. It is possible that this is related to the significantly longer U≡N distances in **3Li** and **3Na** *cf* **3Rb** and **3Cs** together with the planar geometry of the former (Supplementary Note 3).

To provide a unified description of the electronic structure, we have incorporated exchange into our CF + SOC framework for **3Na**. The theory of exchange interactions between orbitally degenerate ions is not trivial[66–68]. One approach, developed to treat interactions between octahedral Co$^{II}$ ($^4$T) ions, is the Lines model[66]. In our case, using a modified Lines approach and assuming an Ising interaction, we have:

$$\hat{H} = \sum_{i=1}^{2}\left[\kappa^2 B'^0_2\hat{O}^0_{2i} + \kappa^4 B'^0_4\hat{O}^0_{4i} + \kappa^6\left(B'^0_6\hat{O}^0_{6i} + B'^6_6\hat{O}^6_{6i}\right)\right. \\ \left. + \kappa\lambda\hat{l}_i\cdot\hat{s}_i + \mu_B\left(\kappa\hat{l}_i + 2\hat{s}_i\right)\cdot B\right] - 2j_{zz}\hat{s}_{z_1}\cdot\hat{s}_{z_2} \quad (2)$$

The uranium ions have the same local parameterizations (determined, and fixed, as above); fitting $j_{zz}$ to the EPR doublet gives $\pm 0.77$ cm$^{-1}$ (Fig. 6e; Table 2). The features at $\sim 1.2$ T are not reproduced, and this did not improve with non-Ising exchange models. An antiferromagnetic interaction would be consistent with the magnetization hysteresis for **3Na**, which collapses at 0.5 T (Fig. 6f), the field at which the non-magnetic ground doublet is crossed by the first excited state (Supplementary Fig. 22).

The first uranium SMMs were reported as recently as 2009 (ref. 69), involving uranium(III), with uranium(V) examples reported later[14,70]. All of **3M–5M**, with the exception of **3Li**, behave as SMMs. Magnetization relaxation times ($\tau$) were determined by fitting $\chi_M''$ data to a generalized Debye model (Fig. 8). The usual interpretation of such data is to extract an effective energy barrier to relaxation ($U_{eff}$) assuming Orbach relaxation, that is, via an excited state by exchange of phonons with the lattice. For this mechanism, $U_{eff}$ can be found from the slope of an Arrhenius plot of $\ln(\tau)$ versus $T^{-1}$ in the high-temperature regime. For **3M–5M**, this gives $U_{eff} = 20–40$ K: the top end of this range (**5Na**) would be among the highest reported for a uranium complex. These values are in the range of the lowest lying excited states determined above, hence Orbach relaxation via this state seems feasible.

However, such Arrhenius plots intrinsically bias the interpretation towards Orbach, because even ideal Raman behaviour can appear to have a linear high-temperature region (Supplementary Fig. 23). Other possible mechanisms[3] include direct relaxation (in a large applied field), Raman relaxation (two-phonon exchange with the lattice) and, at low temperature, quantum tunnelling of magnetization (via near-degenerate states). These mechanisms have different temperature ($T$) and applied magnetic field ($H$) dependences. Simplified expressions are $\tau^{-1} = DH^4T$, $\tau^{-1} = \tau_0^{-1}e^{-U_{eff}/T}$ and $\tau^{-1} = CT^n$, respectively, for direct, Orbach and Raman, while quantum tunnelling of magnetization is temperature independent ($D$ and $C$ are numerical coefficients). In the small d.c. fields and temperature range of our experiments, direct relaxation is negligible. Different presentations of relaxation data can help

distinguish different mechanisms (Supplementary Fig. 23): (i) in an Arrhenius plot, Orbach is linear and Raman is curved and (ii) in a log–log plot of $\tau^{-1}(T)$, these profiles are switched. Examining such plots for **3M–5M** (Fig. 8; Supplementary Fig. 24) shows that Raman dominates. Fitting the data does not require an exponential (Orbach) component, and the power law dependence gives exponents $n = 4$–$7$ (Supplementary Table 17), consistent with Raman relaxation for Kramers ions. Hence, it is not necessary to invoke Orbach relaxation, and there is no correlation of the relaxation to the SO-state energies.

By probing the uranium(V) mono(oxo) and 14 uranium(V) nitride complexes by EPR and UV/Vis/NIR spectroscopies, super-conducting quantum interference device (SQUID) magnetometry, and modelling with CF and *ab initio* calculations, we have quantified a clear difference between the $O^{2-}$ and $N^{3-}$ ligand sets. For the $O^{2-}$ complex **1**, $j_z \approx \pm 3/2$ is clearly the ground state; however, for the $N^{3-}$ complexes **3M–5M**, the ground state is more complex than could be predicted with state-of-the-art computational methods. Modelling all experimental data simultaneously with a CF model has allowed a coherent and detailed picture of the electronic structure of this unique series of complexes, which has implications for our understanding of uranium SMMs and tracing the orbital parentage of SO-coupled states.

## Methods

**General.** All manipulations were performed under a dry, oxygen-free nitrogen atmosphere using Schlenk-line and glove-box techniques, using dried and degassed solvents and reagents. The compounds were characterized by elemental analysis, single crystal X-ray diffraction, nuclear magnetic resonance (NMR), Fourier transform infrared (FTIR), UV/Vis/NIR and EPR spectroscopies, and magnetic studies. See Supplementary Methods for general detail of manipulations and also *ab initio* calculations.

**Preparation of [{U(Tren$^{TIPS}$)(μ-N)(μ-Li)}₂] (3Li).** A solution of **7** (3.57 g, 4.00 mmol) in toluene (10 ml) was added to a cold ( − 78 °C) slurry of Li metal (0.08 g, 11.60 mmol) in toluene (20 ml). The mixture was allowed to slowly warm to room temperature and then stirred for 5 days. Each day the mixture was sonicated for 1 h. After this time, a red precipitate had formed that was extracted with 60 ml hot (100 °C) toluene and filtered through a frit. The residue was washed with hot (100 °C) toluene (2 × 10 ml) and the combined extracts concentrated to ∼ 30 ml and stored at − 30 °C to yield **3Li** as a red crystalline solid. The product was isolated by filtration, washed with pentane (2 × 10 ml) and dried *in vacuo*. Yield 1.22 g, 35%. Anal. calcd for C₆₆H₁₅₀N₁₀Li₂Si₆U₂: C, 45.49; H, 8.68; N, 8.04. Found: C, 45.46; H, 8.62; N, 7.99. ¹H NMR (C₆D₆, 298 K): δ 33.65 (s, 6H, CH₂), 7.84 (s, 6H, CH₂), − 1.33 (s, 9H, CH(CH₃)₂), − 2.74 (s, 54H, CH(CH₃)₂). FTIR ν cm⁻¹ (Nujol): 1,591 (w), 1,302 (w), 1,137 (w), 1,052 (s), 965 (w), 932 (s), 882 (s), 737 (s), 672 (m), 633 (m), 567 (w), 545 (w), 516 (w). ²⁹Si NMR and solution magnetic moment (Evans method) could not be conducted due to the insolubility of **3Li** in aromatic solvent once isolated.

**Preparation of [{U(Tren$^{TIPS}$)(μ-N)(μ-Na)}₂] (3Na).** A solution of **7** (2.68 g, 3.00 mmol) in toluene (10 ml) was added to a cold ( − 78 °C) toluene suspension of Na metal (70 mg, 3.00 mmol). The mixture was allowed to slowly warm to room temperature and then stirred for 5 days. Each day the mixture was sonicated for 1 h. After this time, a red precipitate had formed which was extracted and filtered through a frit using boiling toluene (30 ml). The residue was washed with boiling toluene (2 × 10 ml). The filtrate was concentrated to ∼ 10 ml and stored at − 30 °C to yield **3Na** as a red crystalline solid. The product is isolated by filtration, washed with pentane (2 × 10 ml) and dried *in vacuo*. Yield 1.32 g, 50%. The identity of 3Na was confirmed by comparison with previously reported methods.

**Preparation of [{U(Tren$^{TIPS}$)(μ-N)(μ-K)}₂] (3K).** Toluene (15 ml) was added to a cold ( − 78 °C) stirring mixture of **7** (2.96 g, 3.30 mmol) and KC₈ (0.44 g, 3.30 mmol). The resulting mixture was allowed to warm to ambient temperature and stirred for a further 6 days to afford a dark brown suspension. After this time, the mixture was allowed to thoroughly settle (1 h) and carefully filtered to afford a dark brown solid that was washed with toluene (5 × 5 ml) until the washings obtained were colourless. The solid was then extracted into hot (70 °C) benzene and quickly filtered through a frit to remove the graphite precipitate. The filtrate was stored at 5 °C for 16 hs to yield dark red crystals of **3K** that were isolated by filtration and dried *in vacuo*. Yield: 1.70 g, 57% (crystalline).

Anal. calcd for C₆₆H₁₅₀N₁₀K₂Si₆U₂: C, 43.88; H, 8.37; N, 7.75. Found: C, 44.18; H, 8.38; N, 7.62. FTIR ν cm⁻¹ (Nujol): 1,344 (w), 1,134 (w), 1,069 (s), 1,056 (s), 992 (w), 932 (s), 880 (m), 851 (w), 780 (m), 746 (s), 671 (m), 626 (w), 571 (w), 545 (w), 515 (w). ¹H, ²⁹Si NMR and solution magnetic moment (Evans method) could not be obtained due to the insolubility of **3K** in aromatic solvent once isolated.

**Preparation of [{U(Tren$^{TIPS}$)(μ-N)(μ-Rb)}₂] (3Rb).** A solution of **7** (3.57 g, 4.00 mmol) in toluene (10 ml) was added to a cold ( − 78 °C) slurry of Rb metal (0.38 g, 4.40 mmol) in toluene (10 ml). The mixture was allowed to slowly warm to room temperature and then stirred for 72 h. The resulting mixture was filtered and the remaining solid extracted with toluene (2 × 5 ml). The toluene extracts were combined and volatiles were removed *in vacuo* to afford a red-brown solid. Recrystallization from 10 ml hot (70 °C) benzene afforded **3Rb** as a red crystalline solid on cooling to 5 °C. Yield: 1.90 g, 50% (crystalline). Anal. calcd for C₆₆H₁₅₀N₁₀Rb₂Si₆U₂: C, 41.73; H, 7.96; N, 7.37. Found: C, 39.35; H, 7.29; N, 7.61. FTIR ν cm⁻¹ (Nujol): 1,337 (w), 1,270 (w), 1,135 (m), 1,064 (s), 989 (m), 931 (s), 881 (m), 846 (m), 777 (s), 734 (s), 670 (m), 628 (w), 564 (w), 543 (w), 509 (w). ¹H and ²⁹Si NMR and solution magnetic moment (Evans method) could not be obtained due to the insolubility of **3Rb** in aromatic solvent once isolated.

**Preparation of [{U(Tren$^{TIPS}$)(μ-N)(μ-Cs)}₂] (3Cs).** A solution of **7** (3.43 g, 3.85 mmol) in benzene (10 ml) was added to a cold ( − 78 °C) slurry of Cs metal (0.51 g, 3.85 mmol) in benzene (10 ml). The mixture was allowed to slowly warm to room temperature and then stirred for 72 h. The resulting mixture was filtered and the remaining solid extracted with hot benzene (70 °C, 2 × 5 ml). The benzene extracts were combined and volatiles were removed *in vacuo* to afford a red-brown solid. Recrystallization from hot (70 °C) benzene (10 ml) afforded **3Cs** as a red crystalline solid on cooling to 5 °C. Yield: 1.39 g, 36%. Anal. calcd for C₆₆H₁₅₀Cs₂N₁₀Si₆U₂: C, 39.75; H, 7.58; N, 7.02. Found: C, 40.12; H, 7.37; N, 7.02. ¹H NMR (C₆D₆, 298 K): δ 37.54 (s, 6H, CH₂), 8.19 (s, 6H, CH₂), − 5.14 (s, 9H, CH(CH₃)₂), − 5.64 (s, 54H, CH(CH₃)₂). FTIR ν cm⁻¹ (Nujol): 1,340 (w), 1,251 (w), 1,135 (m), 1,059 (s), 1,056 (s), 990 (m), 932 (s), 880 (m), 843 (m), 772 (s), 739 (s), 670 (m), 626 (w), 597 (m), 566 (w), 542 (w), 511 (w). μeff (Evans method, C₆D₆, 298 K): 1.89 μB. **3Cs** is only partially soluble in aromatic solvents meaning a reliable ¹H NMR could not be obtained. Furthermore, a ²⁹Si NMR spectrum could not be obtained.

**Preparation of [U(Tren$^{TIPS}$)(N)][K(15C5)₂] (4K).** Complex **3K** (1.14 g, 0.63 mmol) was suspended in 15 ml toluene and to it was added a solution of 15-crown-5 (0.56 g, 2.52 mmol) in toluene (10 ml) at room temperature. The red/brown mixture was stirred for a further 16 h and filtered. The solvent was removed *in vacuo* to yield a brown oil that crystallized within 1 h at room temperature. On occasions, storage at 5 °C was necessary to obtain crystalline material. Crystals of **4K** were isolated by thorough washing with pentane (5 × 5 ml) and were dried *in vacuo* to afford a brown crystalline solid. Yield: 1.61 g, 95% (crystalline). Anal. calcd for C₅₃H₁₁₅N₅KO₁₀Si₃U: C 47.36; H 8.63; N 5.21. Found: C 47.47; H 8.44; N 4.93. ¹H NMR (C₆D₆, 298 K): δ 39.10 (s, 6H, CH₂), 11.92 (s, 6H, CH₂), 8.45 (s, 20H, OCH₂), 4.82 (s, 20H, OCH₂), − 5.58 (s, 9H, CH(CH₃)₂), − 6.36 (s, 54H, CH(CH₃)₂). ²⁹Si{¹H} NMR (C₆D₆, 298 K): δ − 15.13. FTIR ν cm⁻¹ (Nujol): 1,302 (m), 1,260 (m), 1,124 (s), 1,042 (m), 935 (m), 882 (m), 856 (m), 741 (m), 723 (m), 670 (w), 625 (w), 565 (m), 506 (w). μeff (Evans method, C₆D₆, 298 K): 1.58 μB.

**Preparation of [U(Tren$^{TIPS}$)(N)][Rb(B15C5)₂] (4Rb).** Toluene (20 ml) was added to a cold ( − 78 °C) mixture of **3Rb** (0.76 g, 0.40 mmol) and benzo-15-crown-5 (0.43 g, 1.60 mmol). The resulting mixture was allowed to warm to room temperature with stirring over 16 h. Volatiles were removed *in vacuo* to afford a brown powder. Recrystallization of the powder from 10 ml warm (60 °C) toluene afforded **4Rb** as a brown crystalline solid on cooling to ambient temperature. Yield: 0.25 g, 21% (crystalline). Anal. calcd for C₆₁H₁₁₅N₅O₁₀RbSi₃U: C, 49.29; H, 7.80; N, 4.71. Found: C, 48.03; H, 7.48; N, 4.64. ¹H NMR (C₆D₆, 298 K): δ 38.62 (s, 6H, CH₂), 11.00–5.00 (br m, 46H, CH₂, OCH₂, Ar-H), − 5.45 (s, 9H, CH(CH₃)₂), − 6.18 (s, 54H, CH(CH₃)₂). ²⁹Si{¹H} NMR (C₆D₆, 298 K): δ − 15.80. FTIR ν cm⁻¹ (Nujol): 1,595 (m), 1,296 (m), 1,217 (m), 1,127 (s), 1,077 (s), 1,042 (m), 934 (m), 882 (m), 851 (w), 739 (s), 670 (m), 627 (m), 542 (w), 507 (w). μeff (Evans method, C₆D₆, 298 K): 2.46 μB.

**Preparation of [U(Tren$^{TIPS}$)(N)][Cs(B15C5)₂] (4Cs).** Toluene (30 ml) was added cold ( − 78 °C) mixture of **3Cs** (0.50 g, 0.25 mmol) and benzo-15-crown-5 (0.27 g, 1.00 mmol). The resulting mixture was allowed to warm to room temperature with stirring over 16 h. Volatiles were removed *in vacuo*, and the product extracted into hot hexanes (60 °C, 10 ml). The mixture was filtered and concentrated to ∼ 2 ml precipitating a sticky solid. The solid was isolated by filtration, suspended in pentane (5 ml) and stored at − 80 °C for 16 h. The pentane was filtered away to afford **4Cs** as a brown-black solid. A concentrated solution of **4Cs** in hexanes afforded brown-black crystals suitable for single crystal X-ray diffraction studies over 48 h. Yield: 0.20 g, 50%. Anal. calcd for

$C_{61}H_{115}CsN_5O_{10}Si_3U$: C, 47.77; H, 7.56; N, 4.57. Found: C, 47.54; H, 7.55; N, 4.56.
$^1$H NMR ($C_6D_6$, 298 K): $\delta$ 38.30 (s, 6H, $CH_2$), 8.66 (s, 6H $CH_2$), 7.80 (s, 4H, Ar-$CH$), 7.43 (s, 4H, Ar-$CH$), 6.50–5.10 (br m, 32H, $OCH_2$), −5.38 (s, 9H, $CH(CH_3)_2$, −6.25 (s, 54H, CH($CH_3)_2$). $^{29}$Si{$^1$H} NMR ($C_6D_6$, 298 K): $\delta$ −15.48 p.p.m.
FTIR $\nu$ cm$^{-1}$ (Nujol): 1,504 (m), 1,409 (w), 1,363 (m), 1,234 (m), 1,221 (m), 1,075 (s), 1,044 (s), 934 (s), 869 (w), 850 (m), 739 (s) 670 (m), 627 (w), 563 (w), 539 (w), 508 (w). $\mu_{\text{eff}}$ (Evans method, $C_6D_6$, 298 K): 2.09 $\mu_B$.

**Preparation of [U(Tren$^{TIPS}$)(μ-N)(μ-Li-12C4)] (5Li).** Toluene (20 ml) was added to a cold (−78 °C) mixture of **3Li** (0.87 g, 0.5 mmol) and 12-crown-4 (0.35 g, 2.00 mmol). The resulting mixture was allowed to warm to room temperature with stirring over 16 h. Volatiles were removed *in vacuo* to afford a sticky brown solid. This solid was then washed with hexanes (5 ml) and filtered to afford **5Li** as a brown powder. Dark red crystals were grown from a concentrated solution of **5Li** in toluene at room temperature. Yield 0.43 g, 41% (crystalline). Anal. calcd for $C_{41}H_{91}N_5LiO_4Si_3U$: C, 47.01; H 8.76; N 6.69. Found: C 45.70; H 8.67; N 6.54. $^1$H NMR ($C_6D_6$, 298 K): $\delta$ 38.61 (s, 6H, $CH_2$), 14.04 (s, 8H, $OCH_2$), 11.28 (s, 8H, $OCH_2$), 7.62 (s, 6H, $CH_2$), −5.55 (s, 9H, $CH(CH_3)_2$), −6.74 (s, 54H, CH($CH_3)_2$). $^{29}$Si{$^1$H} NMR ($C_6D_6$, 298 K): $\delta$ −17.67. FTIR $\nu$ cm$^{-1}$ (Nujol): 1,363 (w), 1,260 (s), 1,135 (s), 1,076 (s), 1,026 (s), 934 (m), 883 (m), 861 (w), 736 (s), 670 (w), 629 (w), 589 (w), 565 (w), 544 (w), 508 (w). $\mu_{\text{eff}}$ (Evans method, $C_6D_6$, 298 K): 2.04 $\mu_B$.

**Preparation of [U(Tren$^{TIPS}$)(μ-N)(μ-K-DB18C6)] (5K).** Toluene (20 ml) was added to a cold (−78 °C) mixture of **3K** (0.61 g, 0.34 mmol) and dibenzo-18-crown-6 (0.24 g, 0.68 mmol). The resulting red/brown mixture was allowed to warm to ambient temperature and stirred at room temperature for a further 16 h. The solvent was removed *in vacuo* to yield a pale brown solid that was extracted with hexanes (4 × 5 ml). The combined extracts were concentrated and stored at 5 °C for 16 h, yielding **5K** as pale brown tablets. Yield: 0.56 g, 66% (crystalline). Anal. calcd for $C_{53}H_{99}KN_5O_6Si_3U$: C, 50.37; H, 7.90; N, 5.54. Found: C, 48.58; H, 7.72; N, 4.85. $^1$H NMR ($C_6D_6$, 298 K): $\delta$ 40.20 (s, 6H, $CH_2$), 9.30 (s, 16H, $OCH_2$), 8.18 (s, 4H, Ar-$H$), 7.91 (s, 6H, $CH_2$), 7.07 (s, 4H, Ar-$H$), −5.88 (s, 9H, $CH(CH_3)_2$), −7.24 (s, 54H, CH($CH_3)_2$). $^{29}$Si{$^1$H} NMR ($C_6D_6$, 298 K): $\delta$ −13.30. FTIR $\nu$ cm$^{-1}$ (Nujol): 1,304 (s), 1,259 (s), 1,233 (m), 1,215 (m), 1,130 (s), 1,092 (m), 1,071 (w), 1,061 (s), 1,024 (m), 959 (w), 949 (w), 934 (m), 882 (w), 857 (w), 740 (s), 723 (s), 670 (w), 627 (w), 588 (w), 566 (w), 514 (w). $\mu_{\text{eff}}$ (Evans method, $C_6D_6$, 298 K): 2.72 $\mu_B$.

**Preparation of [U(Tren$^{TIPS}$)(μ-N)(μ-Rb-18C6)] (5Rb).** Toluene (20 ml) was added to a cold (−78 °C) mixture of **3Rb** (0.66 g, 0.35 mmol) and 18-crown-6 (0.18 g, 0.70 mmol). The resulting mixture was allowed to warm to room temperature with stirring over 16 h. Volatiles were removed *in vacuo* to afford a sticky brown solid. The solid was washed with hexanes (2 × 5 ml) and filtered to afford **5Rb** as a brown powder. Crystalline material was obtained from a concentrated solution of **5Rb** in hexanes. Yield: 0.37 g, 44% (crystalline). Anal. calcd for $C_{45}H_{99}N_5O_6RbSi_3U$: C, 44.52; H, 8.22; N, 5.77. Found: C, 38.36; H, 6.93; N, 7.86. $^1$H NMR ($C_6D_6$, 298 K): $\delta$ 41.91 (s, 6H, $CH_2$), 9.86 (s, 24H, $OCH_2$), 8.27 (s, 6H, $CH_2$), −6.37 (s, 9H, $CH(CH_3)_2$), −7.36 (s, 54H, CH($CH_3)_2$). $^{29}$Si{$^1$H} NMR ($C_6D_6$, 298 K): $\delta$ −11.88. FTIR $\nu$ cm$^{-1}$ (Nujol): 1,352 (s), 1,284 (w), 1,116 (s), 1,079 (m), 1,013 (m), 987 (w), 962 (w), 935 (m), 882 (m), 866 (w), 840 (w), 740 (s), 671 (m), 627 (m), 564 (w), 542 (w), 509 (w). $\mu_{\text{eff}}$ (Evans method, $C_6D_6$, 298 K): 1.89 $\mu_B$.

**Preparation of [U(Tren$^{TIPS}$)(μ-N)(μ-Cs-18C6)] (5Cs).** Toluene (20 ml) was added to a cold (−78 °C) mixture of **3Cs** (0.64 g, 0.32 mmol) and 18-crown-6 (0.17 g, 0.64 mmol). The resulting mixture was allowed to warm to room temperature with stirring over 16 h. Volatiles were removed *in vacuo* and the resulting brown solid washed with hexanes (5 ml). The solid was extracted into toluene (5 ml), concentrated to 2 ml) and stored at −30 °C for 7 days to afford orange crystals of **5Cs**. Yield: 0.14 g, 34%. Anal. calcd for $C_{45}H_{99}CsN_5O_6Si_3U$: C, 42.85; H, 7.91; N, 5.55. Found: C, 42.80; H, 7.84; N, 5.32. $^1$H NMR ($C_6D_6$, 298 K): $\delta$ 40.10 (s, 6H, $CH_2$), 10.89 (s, 12H, $OCH_2$), 7.91 (s, 12H, $OCH_2$), 3.61 (s, 6H, $CH_2$), −5.88 (s, 9H, $CH(CH_3)_2$), −6.71 (s, 54H, CH($CH_3)_2$). $^{29}$Si{$^1$H} NMR ($C_6D_6$, 298 K): $\delta$ −13.32 p.p.m. FTIR $\nu$ cm$^{-1}$ (Nujol): 1,351(m), 1,282 (w), 1,112 (s), 1,082 (m), 1,013 (m), 959 (w), 933 (m), 881 (w), 858 (w), 837 (s), 792 (s), 739 (m), 669 (m), 624 (w), 588 (w), 543 (w), 526 (w), 508 (w). $\mu_{\text{eff}}$ (Evans method, $C_6D_6$, 298 K): 2.20 $\mu_B$.

**CF Hamiltonian and fitting strategy.** The CF Hamiltonian employed here is a modified version of the Stevens operator equivalent formalism, where the operator equivalent factors ($\theta_k$) are subsumed in the crystal field parameters (CFPs) and an orbital reduction parameter ($\kappa^k$) has been appended. The equivalence relations to the other common Stevens formalisms are given below. To use this formalism in PHI, the operator equivalent factors are set to unity. The operator equivalent factors for a single

$f$ electron are: $\theta_2 = -\frac{2}{45}$, $\theta_4 = \frac{2}{495}$ and $\theta_6 = -\frac{4}{3,861}$.

$$\hat{H} = \sum_{k=2,4,6} \sum_{q=-k}^{k} \kappa^k B_k'^q \hat{O}_k^q \equiv \sum_{k=2,4,6} \sum_{q=-k}^{k} \kappa^k B_k^q \theta_k \hat{O}_k^q$$

$$\equiv \sum_{k=2,4,6} \sum_{q=-k}^{k} \kappa^k A_k^q \langle r^k \rangle \theta_k \hat{O}_k^q$$

$$B_k'^q \equiv B_k^q \theta_k \equiv A_k^q \langle r^k \rangle \theta_k$$

Initially, the average orbital energies for the full complexes from *ab initio* calculations were extracted, and the barycentre of each $l_z$ pair was taken. These energies were used to define an initial guess for the axial CFPs through Supplementary Equations 1–3 (Supplementary Note 1). The SOC constant was initially taken as the free-ion value and fixed, while the $B_6'^6$ term was fixed at 1% of $|B_2'^0|$ as described in the text (this ratio was used as a constraint so $B_6'^6$ is subsequently always 1% of $|B_2'^0|$). The axial CFPs were then altered by hand until the correct ground state was obtained (that is, either $j_z \approx \pm 3/2$ or $j_z \approx \pm 5/2$). The relative energies of the excited states as determined by NIR were then fitted by allowing the SOC and CFPs to vary. During this procedure, the lowest two doublets were held in the chosen order by simultaneously fitting their characteristic $g_z$ values (that is, if they switched order during the fitting process the residual would spike sharply). In the event that they did not remain in the correct order, the CFPs were once again manually tweaked to correct the order and the process was repeated until good agreement with the optical data was obtained along with correct ordering of the lowest two states. Then, to achieve the correct $g_z$ value for the $j_z \approx \pm 5/2$ state as measured by EPR, the orbital reduction parameter was 'walked down' from 1 in steps of 0.02, where at each step the SOC and CFPs were allowed to vary to reproduce the optical data. Once a good estimate for the orbital reduction parameter was obtained, all free model parameters (that is, $B_2'^0$, $B_4'^0$, $B_6'^0$, $\lambda$ and $\kappa$) were allowed to vary while simultaneously fitting the NIR, EPR and magnetic data. (EPR linewidths were fixed at those determined with effective $s = 1/2$ models; Supplementary Table 15.) During this process, the simulated susceptibility curve was automatically scaled to match the room temperature moment of the experimental data at each fit iteration, such that the shape of the temperature dependence could be accurately modelled to obtain a reliable measure of the splitting between the lowest two states. This was not performed for the cases where $j_z \approx \pm 5/2$ is the ground state, and such a scaling factor is not included in the final simulations.

**Data availability.** The X-ray crystallographic coordinates for structures reported in this article have been deposited at the Cambridge Crystallographic Data Centre (CCDC), under deposition numbers CCDC 1486679–1486689. These data can be obtained free of charge from The Cambridge Crystallographic Data Centre via www.ccdc.cam.ac.uk/data_request/cif; all other data are available from the authors on request.

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

## Acknowledgements

We thank the Royal Society, the Engineering and Physical Sciences Research Council, the European Research Council, The University of Manchester, The University of Nottingham, the EPSRC UK National EPR Facility, and the National Nuclear Laboratory for supporting this work.

## Author contributions

D.M.K., P.A.C., A.J.W. and B.M.G. synthesized and characterized the compounds. W.L. carried out the X-ray single crystal structure analyses. F.T. collected and analysed magnetometry and EPR data. E.J.L.M. and N.F.C. devised the CF + SOC model. N.F.C. performed Complete Active Space Self-Consistent Field calculations and fitted magnetic and spectroscopic data. E.J.L.M. and S.T.L. originated the central idea, supervised the work and analysed the data. N.F.C., E.J.L.M. and S.T.L. wrote the manuscript with contributions from all co-authors.

## Additional information

**Competing financial interests:** The authors declare no competing financial interests.

