## [Peer Review File · Nature Communications]

Reviewers' comments:

Reviewer #1 (Remarks to the Author):

Review of 100185, Molecular and Electronic Structure of Terminal and Alkali Metal-Capped Uranium(V)-Nitride Complexes

Appropriate number and size of figures: yes.

Appropriate citations: yes, although a few additional citations are suggested below

Recommendation: Return for major revision. The revised paper would be suitable for publication if the issues discussed below can be addressed.

General comments: The topic of this manuscript should be interesting to a wide audience especially to those interested in bonding in inorganic/organometallic systems and those interested in single molecule magnetism. The experimental work and the ab initio calculations in this paper are excellent. The results are clearly communicated. The manuscript is well written and easy to follow. The EPR spectroscopy, modeling, and discussion are especially good. Figure 3 does a great job of clearly communicating the relationship between the strong-field and weak-field models for UN₂⁺ that produces the observable states. The discussion of the EPR spectroscopy and magnetic susceptibility in relation to the ground state is excellent.

The most interesting aspect of this paper is the hybrid approach used to model the electronic structures of actinide complexes. The energies of the excited states are calculated using ab initio methods followed by application of crystal field theory (CFT) using the energies of the excited states to determine the crystal field parameters. While the use of both CFT and theoretical calculations to study the same system has been done numerous times, this hybrid approach is novel. I am aware of only one case in which a similar approach has been used in actinide systems (Notter & Bolvin, J Chem Phys 2009, 130, 184310); however, that paper examined the properties of octahedral, f₁, hexahalide complexes.

If successful, the hybrid approach offers a new, possibly simpler, procedure for modeling and understanding the electronic structures of actinides. However, there are a number of problems with the application of CFT to the complexes in the manuscript, which prevents the accurate modeling of the electronic structure. These issues should be straightforward to address and should allow the authors to better model the electronic structures of the molecules discussed in this manuscript. Given the potential of this hybrid approach, the manuscript should be reconsidered for publication in Nature Communications assuming that the problems with the CFT model can be addressed.

Specific comments:

1) Equations S1 through S3 are incorrect; they are missing a term. Briefly, the authors used the terms in the Hamiltonian as the matrix elements; however, they are not equivalent. The matrix elements contain additional terms. Full explanation is given in section 17.2 of Abragam and Bleaney or Edelstein, N.M. "Electronic Structure of f-Block Compounds" in Organometallics of the f-Elements, Marks, T.J. and Fischer, R.D. , eds. D. Reidel Publishing: Dordrecht, 1979.

The correct equations are as follows:

$$-15\alpha B_2^0 - 600 [\beta B_4^0 - 1260\gamma B_6^0] = E_{\pm 2} \quad (S1)$$

$$-24\alpha B_2^0 - 120\beta B_4^0 + 2520 [\gamma B_6^0] = E_{\pm 1} \quad (S2)$$

$$-27 [\alpha B_2^0 + 180\beta B_4^0 - 3780\gamma B_6^0] = E_0 \quad (S3)$$

where α , β , and γ , are $-(2/45)$, $2/(11*45)$, and $-4/(11*13*27)$ as found in Appendix B, Table 18 of Abragam and Bleaney. To get the correct values for B_{kq}, just divide the ones in the manuscript by α , β , and γ as appropriate.

2) In Eq 1, the use of the orbital reduction factors in conjunction the crystal field parameters is incorrect (nonphysical). The "k" in B_{qk} represents the order of the spherical harmonic and is unrelated to orbital reduction. If anything, a decrease in k will lead to an increase in the crystal field parameters due to greater orbital mixing.

3) Eq 1 is not the correct Hamiltonian for the C₃ point group (it is the Hamiltonian for the D₃ point group). The Hamiltonian for the C_{3v} point group (same as C₃) is as follows (see section 5.7 in Abragam and Bleaney, or better Judd, B.R. Proc. R. Soc. 1955, A232, 458):

$$H \approx B_2^0 O_2^0 + B_4^0 O_4^0 + B_4^3 O_4^3 + B_6^0 O_6^0 + B_6^3 O_6^3 + B_6^6 O_6^6 + \lambda k l \hat{s} + \mu_B (k l + 2s) \cdot B \quad (1)$$

4) It is unclear how the authors performed the CF parameterizations of the CF Hamiltonians apart from using Eq S1 through S3 to determine the values of B_{qk} from the ab initio results for UN₂⁺ with no spin orbit coupling. The authors need to describe the procedure in the SI.

5) The quality of the magnetic susceptibility fits is poor (figures S2-S14), but there is a simple reason. The values of B_{qk} are off by a factor of ~10 for the reasons given above. Once the correct values of B_{qk} are used in PHI, the agreement between experiment and model should greatly improve.

Minor issues and suggestions

1) The authors should describe how the EPR spectra and variable temperature magnetic susceptibility measurements (Figures S2 through S13) were fit in the Experimental Section. This is partially discussed in the manuscript, but the detailed approach should be in the Experimental Section.

2) The title of Figure 3 should begin "Simplified diagram of states arising ..." so that it is more clear that some of the interactions are missing.

3) Figure 3 does not do a great job illustrating why the ground states of the complexes are {plus minus}^{3/2} or {plus minus}^{5/2}. Would it be possible to prepare a similar figure from the ab initio calculations of [UN(NH₂)₃(NH₃)]⁻ to show the splitting of the l_z = {plus minus}² and {plus minus}³ states by the crystal field? My concern would be that such a diagram would be too busy and you would lose the clarity of Figure 3. However, such a figure would be useful if could capture the relationships between the strong field model and the intermediate field model with some clarity.

4) p. 11 line 66. The Lines model fully accounts for the interaction between the between the orbital angular momentum and the spin. The exchange coupling of the true spins, \hat{S} , in Lines' model as well as in the compounds in the manuscript is Heisenberg coupling. The anisotropy in the coupling (e.g., j_{zz}) of the effective spins, \hat{S}_z , is due to spin orbit coupling. The Lines model accounts for this anisotropy. It does not account of anisotropy of the coupling itself (e.g., antisymmetric coupling), but antisymmetric coupling is precluded by inversion symmetry for the molecules in this manuscript.

5) Equation 4 needs to be corrected for dipole-dipole coupling to obtain the correct value of j_{zz} .

6) p.3 line 98: "Miss-matching" should be "mismatching"

7) p. 4 line 70: The B_{k0} notation is Wybourne notation. This should be stated in the text since Stevens Operator Equivalents are mentioned. Stevens actually used a different notation for the crystal field parameters.

8) p. line 39 "states derived from $|l_z| = \dots$ " should be replaced by "states derived primarily from $|l_z| = \dots$ "

9) p. 8 line 64. The g_z value for $|\pm 2, \mp 1/2, \pm 3/2\rangle$ is not 2.00. For this state, g_z is < 2 due to SO coupling to the empty $l_z = \pm 1$ state. For this state, g_z is equal to 2.00 only if the SO coupling is equal to zero. The g_z value for $|0, \mp 1/2, \pm 3/2\rangle$ is 2.00.

Reviewer #2 (Remarks to the Author):

The paper by Eric McInnes and co-workers entitled "Molecular and Electronic Structure of Terminal and Alkali Metal-Capped Uranium(V)-Nitride Complexes" is a beautiful piece of work that will perfectly suits the Nature Communications audience. The authors have not only reported a new method that simplifies and improves the synthesis of terminal uranium(V)-nitride complexes, greatly extending the family to all alkali metals, but have also provided a quantitative understanding of their electronic structure and explored their dynamic magnetic properties. The authors have managed to put together extremely thorough EPR and magnetic data, which combined with multireference wavefunction-based calculations including spin-orbit coupling effects, demonstrate that these systems behave as single molecular magnets. The paper provides the long-time missing explanation of the origin of this effect by means of U(V)-U(V) spin-exchange coupling, measured spectroscopically for the first time here, and definitively deserves to be published in Nature Communications.

I have some comments:

The pioneering work of M. Mazzanti (<http://www.nature.com/nchem/journal/v4/n12/abs/nchem.1494.html>) and the paper by R. Caciuffo et al. (<http://journals.aps.org/physrevlett/pdf/10.1103/PhysRevLett.104.197202>) on single molecular magnet with Np(V) trimers, involving super-exchange coupling should be cited.

Perhaps it will be easier for the reader if the top axis in Figures 5 and 6 is given in g-values instead of Teslas.

A figure that gathers all the main results of this work will be beneficial, i.e., a more generalized version of Scheme 1, describing the synthesis, Figure 4 or part of it with an orbital energy diagram, an EPR spectrum and a χ_T curve, along with a susceptibility curve and the UV-vis spectrum.

Out of a curiosity, have the authors considered explicitly including the N2p orbitals in the active space in their [UN]2+ model?

Reviewer #3 (Remarks to the Author):

The manuscript by Little et al describes original research covering the synthesis, characterization, and electronic structure determination of a family of uranium(V) nitrides. Given their discovery of the terminal uranium(VI) nitride in 2012, this work is a logical extension of that, and worthy of study. In this manuscript, the authors explain a general synthetic technique that they use to access such moieties, and claim that they are "removing the restriction" to these nitrides. There is also a lengthy discussion of various experimental spectroscopic techniques the authors use to characterize these materials. Finally, detailed CASSCF calculations accompany the experimental

results to explain spin orbit coupling and other phenomena.

While the synthetic work is certainly important and fascinating, describing three types of uranium nitride families (1. Dinuclear contact ion pairs, 2. Separated ion-pairs, 3. Contact ion pairs). However, one of each of these types have already been reported (Ref 22, Science; Ref 23, Nat. Chem; Ref 25, ACIE) so the impact of this paper does not rest in the new moieties that have been synthesized or the methodology. The description by the authors does make it sound like any chemist can adapt this technique to their system, but something tells me that this is probably very specific to the TREN ligand framework, and probably could not be extended to non-tripodal environments.

Additionally, throughout the paper, the authors discuss where difficulties lay in sample purification. There is no proof given for the purity of each of the compounds that were used in the various spectroscopic measurements. With sensitive compounds, it's sometimes easier to grow a single crystal than it is to actually recrystallize a sample for purity. I hope the authors realize there is no reason to measure spectroscopy on samples of 80% purity. A few examples (but not exhaustive):

- Page 6 "In some samples we observe weak features in the $g \sim 2$ region which we believe are not intrinsic to the complexes."
- Page 3 "Abstractions proceed quantitatively to give 4Na-4Cs as oils, but crystalline yields vary widely (21-95%)." Does quantitatively mean crude/impure yield?
- Page 6 "UV/Vis/NIR electronic absorption spectra of 3M could not be obtained due to insolubility in non-polar solvents..."
- Page 6 "The only outlier in this trend is 5K, which seems to behave more like the 4M series."
- Figure 5 EPR and Squid. The fits do not seem to be very good. As these techniques are very sensitive to diamagnetic / radical impurities, this may say something about the purity of the samples measured (same with figure 6 per). A technique that tolerates diamagnetic impurities, NIR, the fit looks good (based on the chosen scale...) but there are also other small absorbances in the NIR.
- In Figure 7, I'd expect less variation between samples if only the cation is being changed. Especially for the 4M and 5M series. I could believe differences in the 3M series since the cations are much more intimate with the uranium.

Overall, this paper is very dense and difficult for the general chemist to understand, which seems to be the goal of the Nature Communications journal. Thus, it is likely too specialized, and would be better suited for a more focused journal, perhaps even a journal more geared towards physical chemistry. The authors say this spectroscopic/computational approach is a model to allow "a coherent and detailed picture of the electronic structure of this unique series of complexes to be comprehensively examined". This is not a very informative / general approach, but specialized really only to [(TRENtips)U(N)][M] systems. For example, the authors allude to more generic systems throughout the text, such as octahedral UX61- species (Ref 9), which are more applicable and appreciated by the community as a whole. While the overall approach is an interesting one, the way the data is presented is far too technical for such an audience, and there is doubt about the quality of material being used for the spectroscopic analysis.

Response to Referees' Comments

Referee 1:

General comments: The topic of this manuscript should be interesting to a wide audience especially to those interested in bonding in inorganic/organometallic systems and those interested in single molecule magnetism. The experimental work and the ab initio calculations in this paper are excellent. The results are clearly communicated. The manuscript is well written and easy to follow. The EPR spectroscopy, modeling, and discussion are especially good. Figure 3 does a great job of clearly communicating the relationship between the strong-field and weak-field models for UN₂⁺ that produces the observable states. The discussion of the EPR spectroscopy and magnetic susceptibility in relation to the ground state is excellent.

The most interesting aspect of this paper is the hybrid approach used to model the electronic structures of actinide complexes. The energies of the excited states are calculated using ab initio methods followed by application of crystal field theory (CFT) using the energies of the excited states to determine the crystal field parameters. While the use of both CFT and theoretical calculations to study the same system has been done numerous times, this hybrid approach is novel. I am aware of only one case in which a similar approach has been used in actinide systems (Notter & Bolvin, J Chem Phys 2009, 130, 184310); however, that paper examined the properties of octahedral, f₁, hexahalide complexes.

If successful, the hybrid approach offers a new, possibly simpler, procedure for modeling and understanding the electronic structures of actinides. However, there are a number of problems with the application of CFT to the complexes in the manuscript, which prevents the accurate modeling of the electronic structure. These issues should be straightforward to address and should allow the authors to better model the electronic structures of the molecules discussed in this manuscript. Given the potential of this hybrid approach, the manuscript should be reconsidered for publication in Nature Communications assuming that the problems with the CFT model can be addressed.

Specific comments:

1) Equations S1 through S3 are incorrect; they are missing a term. Briefly, the authors used the terms in the Hamiltonian as the matrix elements; however, they are not equivalent. The matrix elements contain additional terms. Full explanation is given in section 17.2 of Abragam and Bleaney or Edelstein, N.M. "Electronic Structure of f-Block Compounds" in Organometallics of the f-Elements, Marks, T.J. and Fischer, R.D., eds. D. Reidel Publishing: Dordrecht, 1979.

The correct equations are as follows:

$$-15\alpha B_2 - 600 \langle B^2 \rangle - 4 - 1260\gamma B_6 = E_{\pm 2} \quad (S1)$$

$$-24\alpha B_2 - 120\beta B_4 + 2520 \langle \gamma B^2 \rangle - 6 = E_{\pm 1} \quad (S2)$$

$$-27 \langle \alpha B^2 \rangle - 2 + 180\beta B_4 - 3780\gamma B_6 = E_0 \quad (S3)$$

where α , β , and γ , are $-(2/45)$, $2/(11*45)$, and $-4/(11*13*27)$ as found in Appendix B, Table 18 of Abragam and Bleaney. To get the correct values for B_{kq} , just divide the ones in the manuscript by α , β , and γ as appropriate.

RESPONSE: The referee is correct that the operator equivalent factors should be included in the Hamiltonian, and therefore in equations S1 – S3, in traditional Stevens CF notation. However, we would like to point out that we are modelling an *effective* CF that is not purely electrostatic in origin (e.g. orbital splitting also arises due to covalency) and therefore it follows that the full Stevens form is out-dated in that context. We employ the elegant formalism of the Stevens CFT (providing facile calculation of the matrix elements) to build our effective CF Hamiltonian with the correct symmetry properties. Thus, we prefer the ‘naked’ CF Hamiltonian where a single parameter subsumes all multiplicative constants and variables, including the α , β and γ operator equivalent factors. To be explicit, we are using the (Extended) Stevens Operators [see ref 63. Rudowicz, C. Transformation relations for the conventional O_k and normalised O'_k Stevens operator equivalents with $k=1$ to 6 and $-k \leq q \leq k$. J. Phys. C Solid State Phys. 18, 1415–1430 (1985)] with the coefficients $B_2^q = \alpha \langle r^2 \rangle A_2^q$, $B_4^q = \beta \langle r^4 \rangle A_4^q$ and $B_6^q = \gamma \langle r^6 \rangle A_6^q$. This clearly was not immediately obvious in the original submission, so we have added additional description in the main manuscript text to make it more explicit.

2) In Eq 1, the use of the orbital reduction factors in conjunction the crystal field parameters is incorrect (nonphysical). The "k" in B_{kq} represents the order of the spherical harmonic and is unrelated to orbital reduction. If anything, a decrease in k will lead to an increase in the crystal field parameters due to greater orbital mixing.

RESPONSE: We think our notation has given rise to a misunderstanding. The multiplicative factors in Equation 1 are κ (Greek kappa, labelling the orbital reduction factor) *not* lower-case k (the rank of the CF operator). Note that while orbital reduction is often just applied to the Zeeman operator, we feel that this is somewhat non-physical and that the orbital angular momentum should be reduced in all aspects of the Hamiltonian. The expressions for the Extended Stevens operators generally contain powers of the orbital angular momentum to the power of k (the rank of the operator) and so the factor κ^k (kappa to the power of lower-case k) is included with each CF term. While this could also have been subsumed by each CFP (we suppose this is generally why they are left out), we wanted to be explicit in this case. We have added text to alert readers of this possible confusion.

3) Eq 1 is not the correct Hamiltonian for the C_3 point group (it is the Hamiltonian for the D_3 point group). The Hamiltonian for the C_{3v} point group (same as C_3) is as follows (see section 5.7 in Abragam and Bleaney, or better Judd, B.R. Proc. R. Soc. 1955, A232, 458):

$$H = B_2^0 O_2^0 + B_4^0 O_4^0 + B_4^3 O_4^3 + B_6^0 O_6^0 + B_6^3 O_6^3 + B_6^6 O_6^6 + \lambda \kappa^2 \langle r^2 \rangle + \mu_B (\kappa^2 + 2\zeta^2) \cdot B \quad (1)$$

RESPONSE: Yes, the full Hamiltonian for C_3 would include $|q| = 3$ terms. (In fact, the C_3 point group actually also includes the $-q$ operators: see the excellent tables in the back of C. Gorller-Walrand and K. Binnemans, in Handbook on the Physics and Chemistry of Rare Earths, Elsevier, 1996, vol. 23.) We already discuss this explicitly in the manuscript, including our reasons for neglecting such terms

(see the paragraph preceding Eq. 1 in the main text). In summary, we have omitted all $|q| = 3$ operators on the basis of perturbation theory. With reference to Figure 3 of the manuscript, the $|q| = 3$ terms mix the lowest-lying $l_z = \pm 3$ states with the $l_z = 0$ state, as well as the $l_z = \pm 2$ states with the $l_z = \mp 1$ states. Due to the strong axial potential of the nitride, these separations are very large (compared to the direct mixing of the degenerate $l_z = \pm 3$ states with each other by the $|q| = 6$ terms) and so perturbation theory tells us the $|q| = 3$ terms would have to be orders of magnitude larger than the $|q| = 6$ terms to have any influence. Therefore, the most minimal model (to avoid over-parameterisation) is to account for the main effects of the trigonal field (i.e. direct mixing of the degenerate $l_z = \pm 3$ states) with the $|q| = 6$ terms. We have also noted in the paper that we set B_6^{-6} to zero by judicious choice of axes.

4) *It is unclear how the authors performed the CF parameterizations of the CF Hamiltonians apart from using Eq S1 through S3 to determine the values of Bqk from the *ab initio* results for UN2+ with no spin orbit coupling. The authors need to describe the procedure in the SI.*

RESPONSE: This was by no means a trivial task and took a long time before a stable approach could be reached - we thank the referee for pointing out that a clearer description is necessary. The following text has been included in the SI. "Initially, the average orbital energies for the full complexes from *ab initio* calculations were extracted, and the barycentre of each l_z pair was taken. These energies were used to define an initial guess for the axial CFPs through Equations S1 – S3. The SOC constant was initially taken as the free-ion value and fixed, while the B_6^6 term was fixed at 1% of $|B_2^0|$ as described in the text (this ratio was used as a constraint so B_6^6 is subsequently always 1% of $|B_2^0|$). The axial CFPs were then altered by hand until the correct ground state was obtained (i.e. either $j_z \approx \pm 3/2$ or $j_z \approx \pm 5/2$). The relative energies of the excited states as determined by NIR were then fitted by allowing the SOC and CFPs to vary. During this procedure, the lowest two doublets were held in the chosen order by simultaneously fitting their characteristic g_z values (i.e. if they switched order during the fitting process the residual would spike sharply). In the event that they did not remain in the correct order, the CFPs were once again manually tweaked to correct the order and the process was repeated until good agreement with the optical data was obtained along with correct ordering of the lowest two states. Then, in order to achieve the correct g_z value for the $j_z \approx \pm 5/2$ state as measured by EPR, the orbital reduction parameter was 'walked down' from 1 in steps of 0.02, where at each step the SOC and CFPs were allowed to vary to reproduce the optical data. Once a good estimate for the orbital reduction parameter was obtained, all free model parameters (i.e. B_2^0 , B_4^0 , B_6^0 , λ and κ) were allowed to vary whilst simultaneously fitting the NIR, EPR and magnetic data. During this process, the simulated susceptibility curve was automatically scaled to match the room temperature moment of the experimental data at each iteration of fit, such that the shape of the temperature dependence could be accurately modelled to obtain a measure of the splitting between the lowest two states. This was not performed for the cases where $j_z \approx \pm 5/2$ is the ground state, and such a scaling factor is not included in the final simulations."

5) *The quality of the magnetic susceptibility fits is poor (figures S2-S14), but there is a simple reason. The values of Bqk are off by a factor of ~ 10 for the reasons given above. Once the correct values of Bqk are used in PHI, the agreement between experiment and model should greatly improve.*

RESPONSE: There would be no change in the simulations if the operator equivalent factors were included in the Hamiltonian/CFPs, for the reasons described above (they are subsumed in the B_k^q ,

i.e. $B_2^0 \hat{O}_2^0 \rightarrow \alpha B_2^{0'} \hat{O}_2^0$ and $B_2^{0'} = B_2^0/\alpha$, etc.). We agree that the fits to magnetic susceptibility data are not as good as one would expect for e.g. "simple" d-block ions. However, *any* fitting of magnetic data for uranium complexes is extremely rare in the literature because it is not a straightforward task for the reasons discussed in the manuscript introduction. As such, we believe the work reported here is the most comprehensive effort for uranium(V) to date and, importantly, ties the modelling to data from several techniques.

Minor issues and suggestions

1) *The authors should describe how the EPR spectra and variable temperature magnetic susceptibility measurements (Figures S2 through S13) were fit in the Experimental Section. This is partially discussed in the manuscript, but the detailed approach should be in the Experimental Section.*

RESPONSE: This has been added to the SI, as explained above.

2) *The title of Figure 3 should begin "Simplified diagram of states arising ..." so that it is more clear that some of the interactions are missing.*

RESPONSE: This has been done.

3) *Figure 3 does not do a great job illustrating why the ground states of the complexes are {plus minus}3/2 or {plus minus}5/2. Would it be possible to prepare a similar figure from the ab initio calculations of [UN(NH2)3(NH3)]- to show the splitting of the $l_z = \{plus\ minus\}2$ and $\{plus\ minus\}3$ states by the crystal field? My concern would be that such a diagram would be too busy and you would lose the clarity of Figure 3. However, such a figure would be useful if could capture the relationships between the strong field model and the intermediate field model with some clarity.*

RESPONSE: A figure showing this has now been added to the SI (new Supplementary Figure 2), and is cited in the main text.

4) *p. 11 line 66. The Lines model fully accounts for the interaction between the between the orbital angular momentum and the spin. The exchange coupling of the true spins, \hat{S} , in Lines' model as well as in the compounds in the manuscript is Heisenberg coupling. The anisotropy in the coupling (e.g., j_{zz}) of the effective spins, \hat{S}_z , is due to spin orbit coupling. The Lines model accounts for this anisotropy. It does not account of anisotropy of the coupling itself (e.g., antisymmetric coupling), but antisymmetric coupling is precluded by inversion symmetry for the molecules in this manuscript.*

RESPONSE: As the Lines model employs a Heisenberg $\hat{S} \cdot \hat{S}$ type exchange operator, it does not explicitly account for spin-other-orbit or orbit-orbit interactions, though we agree with the referee that it does indirectly include this owing to spin-orbit coupling. We have modified the text to reflect this.

5) *Equation 4 needs to be corrected for dipole-dipole coupling to obtain the correct value of j_{zz} .*

RESPONSE: We use this Hamiltonian to directly fit the experimental data, thus the j_{zz} parameter accounts for the combined exchange and dipolar coupling. Owing to the close proximity of the two low-lying states, we do not wish to further complicate this section by removing an effective dipole component. However, as described in the text, the calculated dipolar interaction in the effective spin

$\frac{1}{2}$ model (ca. 0.005 cm^{-1}) is two orders of magnitude smaller than the experimentally determined $|j_{zz}|$ (ca. 0.4 cm^{-1} in that model). Hence, the correction to Eq (4) would be negligible.

6) p.3 line 98: "Miss-matching" should be "mismatching"

RESPONSE: This has been corrected.

7) p. 4 line 70: The Bk_0 notation is Wybourne notation. This should be stated in the text since Stevens Operator Equivalents are mentioned. Stevens actually used a different notation for the crystal field parameters.

RESPONSE: This point has been addressed above.

8) p. line 39 "states derived from $|l_z| = \dots$ " should be replaced by "states derived primarily from $|l_z| = \dots$ "

RESPONSE: This has been corrected.

9) p. 8 line 64. The g_z value for $|\{\text{plus minus}\}2, \mp 1/2, \{\text{plus minus}\}3/2\rangle$ is not 2.00. For this state, g_z is < 2 due to SO coupling to the empty $l_z = \{\text{plus minus}\}1$ state. For this state, g_z is equal to 2.00 only if the SO coupling is equal to zero. The g_z value for $|0, \mp 1/2, \{\text{plus minus}\}3/2\rangle$ is 2.00.

RESPONSE: We are not quite sure we understand the referee's comments here. The pure state $|\pm 2, \mp 1/2, \pm 3/2\rangle$, which only exists in the strong field limit $CF \gg SO$, has a g_z value of 2.00 as found by equating the z-component of the Zeeman term with that for the effective $S = 1/2$ model: $l_z + 2s_z = g_z \frac{1}{2}$, giving $g_z = 2.00$. SOC couples $|\pm 2, \mp 1/2, \pm 3/2\rangle$ to $|\pm 1, \pm 1/2, \pm 3/2\rangle$ which increases g_z from 2.00 (eventually giving $g_z = 2.57$ in the Russell-Saunders limit, i.e. $SO \gg CF$). There is no such state as $|0, \mp 1/2, \pm 3/2\rangle$ ($|l_z, s_z, j_z\rangle$ notation).

Referee 2:

The paper by Eric McInnes and co-workers entitled "Molecular and Electronic Structure of Terminal and Alkali Metal-Capped Uranium(V)-Nitride Complexes" is a beautiful piece of work that will perfectly suits the Nature Communications audience. The authors have not only reported a new method that simplifies and improves the synthesis of terminal uranium(V)-nitride complexes, greatly extending the family to all alkali metals, but have also provided a quantitative understanding of their electronic structure and explored their dynamic magnetic properties. The authors have managed to put together extremely thorough EPR and magnetic data, which combined with multireference wavefunction-based calculations including spin-orbit coupling effects, demonstrate that these systems behave as single molecular magnets. The paper provides the long-time missing explanation of the origin of this effect by means of $U(V) \leftrightarrow U(V)$ spin-exchange coupling, measured spectroscopically for the first time here, and definitively deserves to be published in Nature Communications.

I have some comments:

The pioneering work of M. Mazzanti (<http://www.nature.com/nchem/journal/v4/n12/abs/nchem.1494.html>) and the paper by R. Caciuffo et al. (<http://journals.aps.org.accessdistant.upmc.fr/prl/pdf/10.1103/PhysRevLett.104.197202>) on single molecular magnet with Np(V) trimers, involving super-exchange coupling should be cited.

RESPONSE: Agreed - we have added these references.

Perhaps it will be easier for the reader if the top axis in Figures 5 and 6 is given in g-values instead of Teslas.

RESPONSE: This has been added to Figures 5 and 6.

A figure that gathers all the main results of this work will be beneficial, i.e., a more generalized version of Scheme 1, describing the synthesis, Figure 4 or part of it with an orbital energy diagram, an EPR spectrum and a χT curve, along with a susceptibility curve and the UV-vis spectrum.

RESPONSE: We thank the reviewer for the suggestion and considered it before - in fact this suggestion is rather close to the ToC figure we have proposed. We have considered including something like this in the main text, but feel it would be repeating information in other figures so we would prefer to keep the structure of figures as they are.

Out of a curiosity, have the authors considered explicitly including the N2p orbitals in the active space in their [UN]2+ model?

RESPONSE: The *ab initio* calculations on the small fragments were only intended to show the general features of the system and not to go for the best possible *ab initio* data. This was attempted for **4Na** where we were interested in the resulting SO states (Table S26), however we found that enlarging the active space did not improve agreement with experiment, and so did not pursue this further.

Referee 3:

The manuscript by Liddle et al describes original research covering the synthesis, characterization, and electronic structure determination of a family of uranium(V) nitrides. Given their discovery of the terminal uranium(VI) nitride in 2012, this work is a logical extension of that, and worthy of study. In this manuscript, the authors explain a general synthetic technique that they use to access such moieties, and claim that they are "removing the restriction" to these nitrides. There is also a lengthy discussion of various experimental spectroscopic techniques the authors use to characterize these materials. Finally, detailed CASSCF calculations accompany the experimental results to explain spin orbit coupling and other phenomena.

While the synthetic work is certainly important and fascinating, describing three types of uranium nitride families (1. Dinuclear contact ion pairs, 2. Separated ion-pairs, 3. Contact ion pairs). However, one of each of these types have already been reported (Ref 22, Science; Ref 23, Nat. Chem; Ref 25, ACIE) so the impact of this paper does not rest in the new moieties that have been synthesized or the methodology. The description by the authors does make it sound like any chemist can adapt this technique to their system, but something tells me that this is probably very specific to the TREN ligand framework, and probably could not be extended to non-tripodal environments.

RESPONSE: With respect to this referee we disagree regarding their latter points. Our nitride remains the only terminal example on record after attempts spanning decades to make it worldwide. Despite the breakthrough, exploiting that advance was hindered by the limitations of the method and the current manuscript reports a much better way of preparing such systems. The importance of the new method is underscored by delivery of 14 compounds, encompassing 3 coordination environments of nitride, that has enabled the entire electronic structure study. We therefore suggest this is an important advance because of what it has enabled. It was not our intention to suggest some broad applicability to any chemical environment; we were referring in the sense of our chemistry. We have made modifications to the text to make this clear now. That said, if another research group comes up with a ligand-set combination that can support a terminal nitride they will now know that in principle there is more than one way to install the nitride linkage at uranium. This also begins to build a relationship to the d-block where there are many ways to construct metal-nitride linkages.

Additionally, throughout the paper, the authors discuss where difficulties lay in sample purification. There is no proof given for the purity of each of the compounds that were used in the various spectroscopic measurements. With sensitive compounds, it's sometimes easier to grow a single crystal than it is to actually recrystallize a sample for purity. I hope the authors realize there is no reason to measure spectroscopy on samples of 80% purity. A few examples (but not exhaustive):

RESPONSE: Of course we discuss any issues encountered in purification/isolation steps, as any synthetic section should, and then we give analytical data (in SI) for the isolated, crystalline materials in each case. We performed magnetic/spectroscopic measurements on these crystalline materials that were analysed prior to measurement, and measurements were made on multiple samples. In the SI we clearly state *"For all measurements, doubly recrystallised powdered samples were carefully checked for purity and data reproducibility between several independently prepared batches for each compound examined"* and *"multiple measurements on multiple samples ensure the reliability of the data"*. Where samples proved difficult to handle in the experiments such that we couldn't get reliable data (e.g. in the Raman spectroscopy) we say so in the text.

- *Page 6 "In some samples we observe weak features in the $g \sim 2$ region which we believe are not intrinsic to the complexes."*

RESPONSE: The EPR spectrum of U(V) is very broad (linewidth) and massively anisotropic (spread over a huge magnetic field range, several T) compared to that of an isotropic free-radical-like $s = \frac{1}{2}$ species (spread over <1 mT). This means that any trace amount of such an $s = \frac{1}{2}$, $g = 2.0$ impurity is artificially prominent in an EPR spectrum. Hence the isotropic and very sharp $g = 2.0$ features that we observe in some samples correspond to an insignificant fraction of the total spectral intensity.

- *Page 3 "Abstractions proceed quantitatively to give 4Na-4Cs as oils, but crystalline yields vary widely (21-95%)."* Does quantitatively mean crude/impure yield?

RESPONSE: We mean that if you do the reaction and take an NMR the starting material is quantitatively converted to the product. The isolated, crystalline materials are analytically pure, whatever the yield of this crystalline product.

• Page 6 "UV/Vis/NIR electronic absorption spectra of 3M could not be obtained due to insolubility in non-polar solvents..."

RESPONSE: We are not sure what is being queried here since it has nothing to do with sample purification. The 3M series aren't soluble so we can't measure their solution optical data.

• Page 6 "The only outlier in this trend is 5K, which seems to behave more like the 4M series."

RESPONSE: This is the experimental observation from repeated measurements. We do not know why 5K appears to behave subtly differently from the other 5M series, and have refrained from speculation. However, as we state in the text, these differences are minor compared to the differences between, for example, terminal nitride and oxide complexes and hence are consistent with the nitride family overall.

• Figure 5 EPR and Squid. The fits do not seem to be very good. As these techniques are very sensitive to diamagnetic / radical impurities, this may say something about the purity of the samples measured (same with figure 6 per). A technique that tolerates diamagnetic impurities, NIR, the fit looks good (based on the chosen scale...) but there are also other small absorbances in the NIR.

RESPONSE: There are, of course, difficulties associated with measuring relatively weak magnetic moments, making them susceptible to small errors in, for example, the exact masses of analyte and eicosane, and the diamagnetic corrections for ligands (Pascal constants are only approximate). This will inevitably contribute to some small variations. Nevertheless, we have performed multiple measurements, and it is also obvious from the data when a sample has degraded. As stated above, there is very little fitted magnetic data on uranium complexes in the literature to compare the goodness of fits with. Researchers have generally just reported the experimental data without any fitting to a model. Hence, we believe we have made a significant step forward, despite the fits not being as good as commonly reported for simpler electronic structure problems.

• In Figure 7, I'd expect less variation between samples if only the cation is being changed. Especially for the 4M and 5M series. I could believe differences in the 3M series since the cations are much more intimate with the uranium.

RESPONSE: The cations are intimately involved with the uranium (capping the nitride) in the 5M series also. However, we understand the referee's point. Magnetic moments reported for uranium(V) in the literature spread over a huge range because of the very high sensitivity to the CF environment. For example, room temperature effective magnetic moments reported for U(V) vary between 1.2 – 3.8 μ_B , and low temperature limiting values from 0.7 – 1.7 μ_B (see D. R. Kindra and W. J. Evans, *Chem. Rev.*, 2014, 114, 8865–8882; reference 58 in the revised manuscript). Hence, the variations we observe within each series are actually relatively small, consistent with the similar coordination environments.

Overall, this paper is very dense and difficult for the general chemist to understand, which seems to be the goal of the Nature Communications journal. Thus, it is likely too specialized, and would be better suited for a more focused journal, perhaps even a journal more geared towards physical chemistry. The authors say this spectroscopic/computational approach is a model to allow "a coherent and detailed picture of the electronic structure of this unique series of complexes to be comprehensively examined". This is not a very informative / general approach, but specialized really

only to [(TRENTips)U(N)][M] systems. For example, the authors allude to more generic systems throughout the text, such as octahedral UX_6^- species (Ref 9), which are more applicable and appreciated by the community as a whole. While the overall approach is an interesting one, the way the data is presented is far too technical for such an audience, and there is doubt about the quality of material being used for the spectroscopic analysis.

RESPONSE: As we point out in the introduction this is inherently a complex area, but one of considerable importance, hence the novelty of presenting such a comprehensive analysis. We simply disagree that tackling such a problem means the work is too specialised, but we have amended the opening paragraphs to give a more general introduction and context as requested. We have also substantially rewritten the manuscript, making it shorter and easier to digest. We also disagree with the referee that our approach “is not very informative or general”: the beauty of the *ab initio* approach is that you can apply it to any compound, as is the spin Hamiltonian approach although additional terms will be necessary as symmetry is lowered. Very high symmetry systems with monatomic ligands such as octahedral UX_6^- were and are important for developing and testing theory. However, if the area is to evolve then we have to move to more complex problems – this makes the results more applicable, not less. Hence, in this work we have studied more ‘real world’ systems involving organic N-donor ligands in heteroleptic complexes (in lower, but still relatively high symmetry). We believe this is a sensible and important next step. Finally, we have taken all reasonable steps to ensure the quality of materials and reproducibility of the results, as described above and in the SI.

Reviewers' comments:

Reviewer #1 (Remarks to the Author):

Response to response letter

The authors have some of the comments, but they have not addressed all of them.

Major issues

1) Equations S1 through S3 are incorrect; they are missing a term. Briefly, the authors used the terms in the Hamiltonian as the matrix elements; however, they are not equivalent. The matrix elements contain additional terms. Full explanation is given in section 17.2 of Abragam and Bleaney or Edelstein, N.M. "Electronic Structure of f-Block Compounds" in Organometallics of the f-Elements, Marks, T.J. and Fischer, R.D. , eds. D. Reidel Publishing: Dordrecht, 1979.

The authors can use whatever formula they like for the crystal field parameters, including the approach that they have used in this manuscript. However, the authors are using an established notation for their crystal field parameters – Wybourne notation. If they are going to use this notation, they need to apply the formalism as explained in the initial review. If they want to use another formalism, they must use a different notation (replace "B" in "B_{qk}" by some other letter other than A, C, O, or Y).

2) In Eq 1, the use of the orbital reduction factors in conjunction the crystal field parameters is incorrect (nonphysical).

Addressed by the authors.

3) Eq 1 is not the correct Hamiltonian for the C₃ point group (it is the Hamiltonian for the D₃ point group). The Hamiltonian for the C_{3v} point group (same as C₃) is as follows (see section 5.7 in Abragam and Bleaney, or better Judd, B.R. Proc. R. Soc. 1955, A232, 458):

While I agree that the authors have explained why they use the D₃ point group rather than the correct C₃ point group, they must refer to the symmetry as "D₃." The operators in eq 1 are for the D₃ point group. Please change the "C₃" in the text to "D₃" when referring to this Hamiltonian.

4) It is unclear how the authors performed the CF parameterizations of the CF Hamiltonians apart from using Eq S1 through S3 to determine the values of B_{qk} from the ab initio results for UN₂₊ with no spin orbit coupling. The authors need to describe the procedure in the SI.

Addressed by the authors.

5) The quality of the magnetic susceptibility fits is poor (figures S2-S14), but there is a simple reason. The values of B_{qk} are off by a factor of ~10 for the reasons given above. Once the correct values of B_{qk} are used in PHI, the agreement between experiment and model should greatly improve.

I am somewhat confused by the authors' response. From the reference 60, I assume the authors are using the program PHI to calculate the magnetic susceptibilities, which makes sense as one of the authors is also the author of PHI. From its Users' Guide, PHI calculates magnetic susceptibility using the conventional crystal field parameters (Wybourne's notation), which are an order of magnitude larger than the parameters in Table 2. In this case, the energies of the excited states calculated using PHI will be much lower than the authors expect.

However, if the authors are not using PHI and are calculating the magnetic susceptibility using

their own code, this is not an issue for the reasons explained in the response letter. In that case, please remove the reference to PHI and describe how the magnetic susceptibility is calculated in the SI. There is currently no information about calculation of the magnetic susceptibility in either the manuscript or SI.

All minor issues addressed except for

4) p. 11 line 66. The Lines model fully accounts for the interaction between the between the orbital angular momentum and the spin. The exchange coupling of the true spins, \hat{S} , in Lines' model as well as in the compounds in the manuscript is Heisenberg coupling. The anisotropy in the coupling (e.g., j_{zz}) of the effective spins, \hat{S}_z , is due to spin orbit coupling. The Lines model accounts for this anisotropy. It does not account of anisotropy of the coupling itself (e.g., antisymmetric coupling), but antisymmetric coupling is precluded by inversion symmetry for the molecules in this manuscript.

The manuscript does not seem to have been changed. It still incorrectly states that the Lines model "neglects any explicit interactions involving the orbital angular momenta."

Reviewer #2 (Remarks to the Author):

Accept the revised manuscript.

Reviewer #3 (Remarks to the Author):

We agree that the author has attempted to generalize the work, and the efforts are appreciated. While the work is still explained to the specialist, it seems that the authors have addressed/responded to many of the concerns that were originally raised.

As an aside, it is important to note that the previous review was in no way an attempt to detract from the importance of the terminal nitride moiety; however, this has been presented in multiple forums and the synthetic work is not the focus of this manuscript. The only role the synthetic work plays in this study is preparation of the analytically and spectroscopically pure samples. All of the data interpretation central to the paper relies on this sample preparation, and the authors have made their case.

Based on these improvements and clarifications, publication is warranted at this time.

Response to Referees' Comments

Reviewer #1 (Remarks to the Author):

Response to response letter

The authors have some of the comments, but they have not addressed all of them.

Major issues

1) Equations S1 through S3 are incorrect; they are missing a term. Briefly, the authors used the terms in the Hamiltonian as the matrix elements; however, they are not equivalent. The matrix elements contain additional terms. Full explanation is given in section 17.2 of *Abragam and Bleaney or Edelstein, N.M. "Electronic Structure of f-Block Compounds" in Organometallics of the f-Elements, Marks, T.J. and Fischer, R.D. , eds. D. Reidel Publishing: Dordrecht, 1979.*

The authors can use whatever formula they like for the crystal field parameters, including the approach that they have used in this manuscript. However, the authors are using an established notation for their crystal field parameters – Wybourne notation. If they are going to use this notation, they need to apply the formalism as explained in the initial review. If they want to use another formalism, they must use a different notation (replace "B" in "Bkq" by some other letter other than A, C, O, or Y).

RESPONSE: We have renamed the Bkq parameters as B'kq, and explicitly address the correspondence to the Stevens formalism, with operator equivalent factors, in a new note in the Supplementary Methods (Crystal field Hamiltonian and fitting strategy). The latter is cited explicitly in the main text where the crystal field equations are introduced.

2) In Eq 1, the use of the orbital reduction factors in conjunction the crystal field parameters is incorrect (nonphysical).

Addressed by the authors.

3) Eq 1 is not the correct Hamiltonian for the C3 point group (it is the Hamiltonian for the D3 point group). The Hamiltonian for the C3v point group (same as C3) is as follows (see section 5.7 in *Abragam and Bleaney, or better Judd, B.R. Proc. R. Soc. 1955, A232, 458*):

While I agree that the authors have explained why they use the D3 point group rather than the correct C3 point group, they must refer to the symmetry as "D3." The operators in eq 1 are for the D3 point group. Please change the "C3" in the text to "D3" when referring to this Hamiltonian.

RESPONSE: We have altered the text to be clear that this is the D_3 Hamiltonian.

4) It is unclear how the authors performed the CF parameterizations of the CF Hamiltonians apart from using Eq S1 through S3 to determine the values of Bqk from the ab initio results for UN2+ with no spin orbit coupling. The authors need to describe the procedure in the SI.

Addressed by the authors.

5) *The quality of the magnetic susceptibility fits is poor (figures S2-S14), but there is a simple reason. The values of B_{qk} are off by a factor of ~ 10 for the reasons given above. Once the correct values of B_{qk} are used in PHI, the agreement between experiment and model should greatly improve.*

I am somewhat confused by the authors' response. From the reference 60, I assume the authors are using the program PHI to calculate the magnetic susceptibilities, which makes sense as one of the authors is also the author of PHI. From its Users' Guide, PHI calculates magnetic susceptibility using the conventional crystal field parameters (Wybourne's notation), which are an order of magnitude larger than the parameters in Table 2. In this case, the energies of the excited states calculated using PHI will be much lower than the authors expect.

However, if the authors are not using PHI and are calculating the magnetic susceptibility using their own code, this is not an issue for the reasons explained in the response letter. In that case, please remove the reference to PHI and describe how the magnetic susceptibility is calculated in the SI. There is currently no information about calculation of the magnetic susceptibility in either the manuscript or SI.

RESPONSE: The confusion comes down to the Crystal Field formalism: this has been addressed in the new text, definitions and equations in the Supplementary Methods (Crystal field Hamiltonian and fitting strategy), as noted above. Yes, PHI is used for the calculation of all magnetic properties (now explicitly stated in the main text, where ref. 60 is cited). [Note that in PHI the operator equivalent factors can be set to unity in order to use our effective CF Hamiltonian.]

All minor issues addressed except for

4) *p. 11 line 66. The Lines model fully accounts for the interaction between the between the orbital angular momentum and the spin. The exchange coupling of the true spins, \hat{S} , in Lines' model as well as in the compounds in the manuscript is Heisenberg coupling. The anisotropy in the coupling (e.g., j_{zz}) of the effective spins, \hat{S}_z , is due to spin orbit coupling. The Lines model accounts for this anisotropy. It does not account of anisotropy of the coupling itself (e.g., antisymmetric coupling), but antisymmetric coupling is precluded by inversion symmetry for the molecules in this manuscript.*

The manuscript does not seem to have been changed. It still incorrectly states that the Lines model "neglects any explicit interactions involving the orbital angular momenta."

RESPONSE: We have amended the text (and removed this statement) to avoid confusion.

Reviewer #2 (Remarks to the Author):

Accept the revised manuscript.

Reviewer #3 (Remarks to the Author):

We agree that the author has attempted to generalize the work, and the efforts are appreciated. While the work is still explained to the specialist, it seems that the authors have addressed/responded to many of the concerns that were originally raised.

As an aside, it is important to note that the previous review was in no way an attempt to detract from the importance of the terminal nitride moiety; however, this has been presented in multiple forums and the synthetic work is not the focus of this manuscript. The only role the synthetic work plays in this study is preparation of the analytically and spectroscopically pure samples. All of the data interpretation central to the paper relies on this sample preparation, and the authors have made their case.

Based on these improvements and clarifications, publication is warranted at this time.